# Diversity and structural differences of bacterial microbial communities in rhizocompartments of desert leguminous plants

Ziyuan Zhou[1,2], Minghan Yu[1,2]*, Guodong Ding[1,2]*, Guanglei Gao[1,2], Yingying He[1,2]

**1** Yanchi Research Station, School of Soil and Water Conservation, Beijing Forestry University, Beijing, P. R. China, **2** Key Laboratory of State Forestry Administration on Soil and Water Conservation, Beijing Forestry University, Beijing, P. R. China

\* yuminghan@bjfu.edu.cn (MY); dingguodong@bjfu.edu.cn (GD)

**Data Availability Statement:** All relevant data are uploaded to the Dryad repository and publicly accessible via the following URL: https://doi.org/10.5061/dryad.9cnp5hqf0.

## Abstract

By assessing diversity variations of bacterial communities under different rhizocompartment types (i.e., roots, rhizosphere soil, root zone soil, and inter-shrub bulk soil), we explore the structural difference of bacterial communities in different root microenvironments under desert leguminous plant shrubs. Results will enable the influence of niche differentiation of plant roots and root soil on the structural stability of bacterial communities under three desert leguminous plant shrubs to be examined. High-throughput 16S rRNA genome sequencing was used to characterize diversity and structural differences of bacterial microbes in the rhizocompartments of three xeric leguminous plants. Results from this study confirm previous findings relating to niche differentiation in rhizocompartments under related shrubs, and they demonstrate that diversity and structural composition of bacterial communities have significant hierarchical differences across four rhizocompartment types under leguminous plant shrubs. Desert leguminous plants showed significant hierarchical filtration and enrichment of the specific bacterial microbiome across different rhizocompartments ($P < 0.05$). The dominant bacterial microbiome responsible for the differences in microbial community structure and composition across different niches of desert leguminous plants mainly consisted of Proteobacteria, Actinobacteria, and Bacteroidetes. All soil factors of rhizosphere and root zone soils, except for $NO_3^-N$ and TP under *C. microphylla* and the two *Hedysarum spp.*, recorded significant differences ($P < 0.05$). Moreover, soil physicochemical factors have a significant impact on driving the differentiation of bacterial communities under desert leguminous plant shrubs. By investigating the influence of niches on the structural difference of soil bacterial communities with the differentiation of rhizocompartments under desert leguminous plant shrubs, we provide data support for the identification of dominant bacteria and future preparation of inocula, and provide a foundation for further study of the host plants-microbial interactions.

**Funding:** This study was supported by the National Key Research and Development Program of China (no. 2016YFC0500905 and 2018YFC0507102), and the National Natural Science Foundation of China (no. 31270749).

**Competing interests:** The authors have declared that no competing interests exist.

## Introduction

Soil microbes involved in soil carbon and nitrogen cycling exert a notable influence on global climate change [1]. The effect of global climate change on desert ecosystems is noticeable [2], resulting in spatial, large-scale variations in desert vegetation communities [3]. Frequent anthropogenic interferences and intensified global climate change, as well as other influencing factors, is accelerating vegetation degradation in arid, semi-arid, and dry sub-humid regions. Vegetation degradation in these areas will alter the regional ecological balance, resulting in land degradation and desertification becoming important areas of concern [4].

Vegetation restoration practices have been extensively undertaken in northwest China since the 1980s, being one of the most effective and sustainable means of controlling desertification and restoring degraded land [5]. Plants such as *Hedysarum mongolicum*, *H. scoparium*, *C. microphylla*, and *Artemisia ordosica* are highly tolerant to arid and areas susceptible to wind-erosion, and they are believed to be suitable sand-fixing plants. In the Mu Us Desert in northwest China, these xeric shrub species are dominant plant community species [6–8], with the majority being leguminous plants.

Legumes have high species diversity and wide tolerance in various global ecosystems [9, 10]; these superiorities are largely due to the unique symbiotic interaction between rhizobia and legumes [11–13]. The symbiosis between legumes and rhizobia is particularly important for biological nitrogen fixation. It not only provides important biological nitrogen source to agriculture and natural ecosystems, but also is conducive to the sustainable production of agriculture by providing necessary ecological services that are of great significance for increasing soil fertility [14]. Many leguminous species also have important economic and ecological value, such as being sources of food, feed, materials, medicinal ingredients, and ornamental plants [14–16].

In desert environments with extreme climates, vegetation growth plays a vital role in wind prevention and sand fixation; in particular, well-developed and widely distributed plant roots help to fix and improve soil quality [7]. Soil not only supports plant growth, it also provides nutrients required for plant growth. In turn, plants can fix external carbon sources, thereby contributing to the improvement of physical, chemical, and biological soil properties [17]. Soil microbes also play a critical role in the complicated interactions between plants and soil [18, 19]. It has been shown that the plasticity of desert vegetation is, to a very large extent, attributable to the influence of soil microbes [20, 21]. Rhizobia inoculation and organic improvers are widely used in the vegetation restoration and soil quality improvement of the degraded Mediterranean area [22–25]. The addition of bacteria can increase the proliferation and development of natural microbes in soil in arid regions, and can subsequently promote soil enzyme activities [26] and increase a supply of low-mobility nutrients, such as potassium and phosphorus, in the soil, which can in turn promote plant growth [27, 28]. Soil microbes participate in and control many soil ecosystem functioning processes, and they are important for maintaining the sustainable development and stability of soil ecosystem functions.

The plant microbiome is composed of different microbe classes, such as bacteria, archaea, oomycetes, fungi, and viruses, and it is often considered to be the second or extended genome of host plants [29]. The microbiome not only provides functional assistance and support to host plants, it also plays a vital role in regulating the health and plasticity of individual plants and the productivity of plant communities [30–34]. The role of rhizobia has always been the main research focus, and its most notable feature lies in its ability to establish a compatible interaction with molecules in the root of the host plant. This interaction depends on the expression of symbiosis-specific genes involving in gene compiling and signal output, and genes that can eventually trigger the unique organogenesis and other corresponding

physiological responses of the host plants [35]. One study has shown that *Rhizobium sullae* is a highly efficient nitrogen-fixing rhizobium that can form symbiotic associations with *Hedysarum* plants [36]. The bacteria cannot survive under extreme conditions to which *Hedysarum* plants can tolerate; however, it can survive in the host plants that are exposed to drought or alkaline conditions [35]. A study on genetic diversity of the rhizobium in plants of *Caragana* Fabr. in the Mu Us Desert has revealed that when there is a shortage of water and nitrogen, the rhizobium may be considered as the basic functional unit of the local ecosystem [37]. *Rhizobium alkalisoli* isolated from *Caragana* plants may promote the growth and environmental tolerance of the plants grown in saline-alkali arid areas [38]. In turn, plant communities exert an influence on soil microbial communities through soil nutrient cycling and other ecological processes [17]. For example, most soil microflorae are carbon-starved [39]. Plants exude up to 40% of photosynthates to rhizospheres [40], resulting in the microbial community density of rhizosphere soil to undergo significant differentiation relative to that of bulk soil. Rhizospheres are also the areas where plants and soil microbes engage in the most intense interactions with each other [41].

Rhizospheres are narrow soil areas affected by root exudates, where up to $10^{11}$ microbial cells can be present [42] and there can be more than 33,000 bacterial and archaeal species per gram of root [28]. Plant roots use bulk soil as a microbial diversity pool in which they can induce the enrichment of specific microbes favorable for plant growth [32]. In general, the number of species of microbial communities in rhizospheres is lower than that in bulk soil [43–45]. Significant differences in bacterial microbial communities between rhizosphere soil and non-rhizosphere soil is referred to as the "rhizosphere effect" [46]. This phenomenon suggests that soil type is an important driver of microbial community composition in the rhizosphere [47, 48]. As far as rhizocompartments are concerned as special micro-ecosystems, the relative differentiation of bacterial communities driven by various soil factors in roots, rhizosphere soil, and non-rhizosphere soil can be used to reflect the intensity of the "rhizosphere effect", and characterize the degree of interactions between plants and soil. This method provides the opportunity to quantitatively examine bacterial microbial community diversity and structural differentiation in rhizocompartments of desert leguminous plants. The main hypotheses of this study, therefore, are: (i) Diversity and structural compositions of soil microbial communities present hierarchical differences across the four rhizocompartments (roots, rhizosphere soil, root zone soil, and inter-shrub bulk soil); (ii) Desert leguminous plants have a hierarchical filtering and enriching effect on the specific beneficial microbes in soil via rhizocompartments.

## Materials and methods

### Research site

This study was undertaken at the Yanchi Research Station (37°04′-38°10′ N and 106°30′-107°47′ E) in Ningxia Province, northwest China, during September 2018. The site, located on the southern fringe of the Mu Us Desert, is characterized by a typical semi-arid continental monsoon climate that is dry and warm throughout the year. The study area has a mean annual temperature of 8.1°C (daily temperature range: -29.4°C ~37.5°C; relative humidity range: 49% ~55%), and a mean annual rainfall around 292 mm, 60%~70% of which falls between July and September (maximum in August), the local soil type is eolian sandy soil [49]. Soil texture in the upper 1 m soil profile is classed as being sandy, having a mean bulk density of 1.5 g cm$^{-3}$ [7]. Ecological degradation in this area is mainly caused by overgrazing, climate change, and wind erosion, resulting in the degradation of arid grasslands into sandy land. Existing vegetation was established via aerial seeding (*A. ordosica*, *H. mongolicum*, and *H. scoparium*),

seedling planting (*C. microphylla*), and natural restoration [7], all of which has been undertaken since 2001. Currently, dominant xeric shrub species in this area includes *A. ordosica*, *H. mongolicum*, *H. scoparium*, *C. microphylla*, and *Salix psammophila*.

## Sampling strategy and sample preparation

In this study, we selected three common local dominant xeric leguminous shrub species, namely *Caragana microphylla*, *Hedysa.rum mongolicum*, and *Hedysarum scoparium*, and studied the changes of bacterial community diversity in four rhizocompartments of their microbiota, including three types of under-shrub samples (i.e., root, rhizosphere soil, root zone soil) and inter-shrub bulk soil.These leguminous plants naturally coexist and they are widely distributed in the desert areas of northwest China (including the study area). The tolerance of these plant species to desert environments and their ecological restoration effects have previously received close attention [6–8]. Samples used in this study were collected during the ripening period of test plants in September 2018. Sample plants were mainly distributed on sunny slopes of fixed dunes formed via natural restoration, and each plant species was sampled with three sample plots (100×100 m, with a spacing of 100 m). Since the sampling site was a natural enclosure, plants such as *Artemisia ordosica*, *Setaria viridis*, and *Salix psammophila* were sparsely distributed in the site. Thus, in this study, the interference of these plants (> 5 m away) was avoided during sampling. For each shrub species, five shrubs were randomly collected from each sample plot to collect root and soil samples. Test samples were composed of the four rhizocompartments of leguminous plants: roots, rhizosphere soil, root zone soil, and inter-shrub bulk soil. Based on local conditions, sampling methods described by Beckers et al. and Xiao et al. were followed when sampling the four rhizocompartment types [50, 51]. During sampling, plant roots were exposed and removed. Root zone soil, consisting of blocky soil by shaking and kneading from the root samples, was collected and stored in sterile sample bags. Soil particles adhering to the roots were collected using tweezers, was identified as rhizosphere soil only to determine soil physicochemical properties. Inter-shrub bulk soil collected in this study is the soil between two adjacent shrubs (about 2 m apart), where plants do not grow, and its sampling depth was the same as that of the root zone soil. Root samples were collected from secondary or tertiary root branches without nodules. Healthy and intact root samples (5–8 cm in length) with similar diameters were cut and stored in sterile sample bags to avoid contaminations from nodule endophytes and other foreign bacteria. All sampling tools (tweezers, blades, etc.) were wiped and sterilized with alcohol (75%) after each use to prevent contamination of the samples from foreign bacteria.

Root samples were initially oscillated at maximum speed for 10 min in 50 mL centrifuge tubes containing 25 mL Phosphate buffer saline (PBS; 130 mM NaCl, 7 mM $Na_2HPO_4$, 3 mM $NaH_2PO_4$, pH 7.0, 0.02% Silwet L-77). The new PBS buffer was replaced during each repetition, and the time interval between the two operations was 5 minutes. The turbid liquid then filtered through a 100-μm nylon mesh cell strainer, and the remaining liquid was centrifuged for 15 min at 3,200g to form a pellet, which containing fine sediment and microorganisms as the rhizosphere soil (namely, microbes that are strongly adhered to the root-associated soil). Added 25 ml sterile PBS buffer into a new sterile 50 ml tube which contented the aforementioned root samples and vortexed, repeat this step until the PBS buffer was clear. The washed roots were placed into new centrifuge tubes containing 25 mL PBS buffer for ultrasonic oscillation for 5min; an interval of 30 s was used between the two replicates. After discarding the fluid, the clean root samples were collected. Xiao et al. reported that the method for cleaning root tissues prevented contamination and damage to plant tissue samples, thereby guaranteeing the ability of treated samples to represent the rhizocompartmental properties of examined

shrub species [51]. We followed the method described by Sun et al. to mix samples for the analysis of bacterial community diversity [8], each collected plant species has three replicates corresponding to its three sample plots. The samples from the rhizocompartments of the five shrubs in the same sample plot were mixed, and these mixed samples were established corresponding to the four rhizocompartment types of each plant species. Each mixed soil sample was homogenized and filtered through a 2 mm soil sieve to remove gravels and impurities. Each mixed sample was divided into two subsamples: Part 1 contained 36 DNA samples (3 shrub species × 4 rhizocompartment types × 3 replicates), which were stored at -80°C for molecular biological analysis; Part 2 contained 27 soil samples (3 shrub species × 3 rhizocompartment types × 3 replicates), which were air-dried for physicochemical analysis.

In order to determine soil physicochemical properties, a conventional approach was adopted to quantify total soil organic carbon (SOC), total nitrogen (TN), total phosphorus (TP), available phosphorus (AP), ammonium nitrogen ($NH_4^+$-N), nitrate nitrogen ($NO_3^-$N), and pH. SOC was quantified using the dichromate oxidation method [52], TN concentration of soil samples was determined using a semi-micro Kjeldahl Apparatus Nitrogen Autoanalyzer [53]. AP and TP were measured using an ultraviolet spectrophotometer (UV-2550; Shimadzu, Kyoto, Japan); $NH_4^+$-N and $NO_3^-$N were measured using the indophenol blue method and the hydrazine sulphate method, respectively; and soil pH was recorded on a 1:1 (10 g:10 mL) soil/distilled water slurry; all these methods were analyzed following the international standard methods as adopted and published by the Institute of Soil Science, Chinese Academy of Sciences (1978). Soil water content (SWC) was measured using a portable soil moisture meter (TRIME-PICO64/32; IMKO, Ettlingen, Germany). When the rhizocompartment was dug and sampled, the SWC was rapidly measured prior to the collection of corresponding samples. The portable soil moisture meter that was used at the time has a probe that is too large to accurately detect the soil water content (SWC) in a very small area of rhizosphere. In addition, the rhizosphere soil that was subsequently cleaned and separated by PBS buffer could not suitably be used for the measurement of soil water content, as the results may not be accurate. Therefore, the measured value of the soil in the root zone was only used as the soil moisture content shared by the rhizosphere and the root zone in the subsequent analysis and calculation. Although the physical and chemical properties of root tissue were not measured, the subsequent analysis showed that rhizosphere soil associated endophytic bacteria communities were most closely interact with the physical and chemical properties of the rhizosphere soil, which is an indication that the two are correlated.

## 16S rRNA genome sequencing and bioinformatics analysis

The DNA extraction processes of plant tissue and soil samples were carried out in strict accordance with the manufacturer's instructions. Before DNA extraction, the plant root samples were frozen in liquid nitrogen and crushed in a mortar. The extraction of DNA from all samples (0.5 g each) presented in this study was carried out using E.Z.N.A. soil DNA kits (OMEGA, United States). The extracted genomic DNA was stored at -80°C until subsequent use. After thawing on ice, extracted DNA samples were separately centrifuged and fully mixed; sample quality was determined using a NanoDrop instrument, and 30 ng DNA was used for PCR amplification. PCR amplification was performed in 25 μL reaction volumes containing 10×PCR buffer, 0.5 μL dNTPs, 1 μL of each primer, 3 μL bovine serum albumin (2 ng/μL), 12.5 μL KAPA 2G Robust Hot Start Ready Mix, ultrapure $H_2O$, and 30 ng template DNA. The PCR amplification program included initial denaturation at 95°C for 5 min, followed by 25 cycles of denaturing at 95°C for 45 s, annealing at 55°C for 50 s, and extension at 72°C for 45 s. Finally, the PCR amplification program was completed at 4°C. Forward primer F799 (5′−

AACMGGATTAGATACCCKG–3′) and reverse primer R1193 (5′–ACGTCATCCCCACCTTCC–3′) were used to target the V5-V7 regions of 16S rRNA. Studies have shown that using the variable regions V5–V7 [54–56] as the sequenced samples can more effectively reduce host contamination compared to using the V3–V4 regions [57–59]. Both primers contained Illumina adapters, and the forward primer contained an 8 bp barcode sequence unique to each sample. An Agarose Gel DNA purification kit (Axygen Biosciences, Union City, CA, the USA) was used for the purification and combination of PCR amplicons. After purification, PCR amplicons were mixed at an equal molar concentration, followed by pair-end sequencing using the Illumina MiSeq sequencing system (Illumina, the USA) according to a standardized process.

MiseqPE300 (Illumina, the USA) sequencing results were recorded in the Fastq format. Quantitative insights into microbial ecology software (QIIME; Version 1.8 http://qiime.org/) was used to analyze original Fastq files and to undertake quality control according to the following criteria [60, 61]: (i) base sequences with a quality score <20 at read tails were removed, and the window was set at 50 bp. When the mean quality score in the window was <20, posterior-end base sequences were discarded from the window and reads shorter than 50 bp were removed after quality control; (ii) paired reads were assembled into one sequence according to the overlapping relationship between reads (minimum overlapping length: 10 bp); (iii) the maximum allowable mismatch ratio of the overlapping areas of assembled sequences was set to 0.1, and sequences failing to meet this criterion in pairs were removed; (iv) the samples were distinguished by the barcodes and the primers that are complementary to both ends of the sequence, and the number of mismatches allowed by the barcode was 0; and (v) different reference databases were selected according to the type of sequencing data (Silva-SSU128-16S rRNA database-bacteria, https://www.arb-silva.de). Chimeras were removed using the Usearch program V8.1861 (http://www.drive5.com/usearch/), and clean tags of high-quality sequences were acquired after smaller-length tags were discarded using Mothur V1.30.1, (http://www.mothur.org/wiki/MiSeq_SOP) [62]. Sequences were clustered into Operational Taxonomic Units (OTUs) using UPARSE V7.1 (http://drive5.com/uparse/) based on a 97% sequence similarity cutoff (excluding single sequences). Representative sequences and an OTU table were obtained [63].

## Statistics analysis

Differences among treatments for diversity index, relative abundance data at phylum-order-genus levels and soil physicochemical factors were analyzed using one-way ANOVA model incorporating shrub species, plant rhizocompartments, and their interaction as fixed factors. Post-hoc comparison LSD tests were performed at the confidence level of 0.05. The relative abundance data of all the major contributing bacteria taxa filtered from the four rhizocompartments at phylum-order-genus levels were log-transformed, thereby adhering to the requirements for normality of data and homogeneity of variance. The Shapiro-Wilk test and the Levene test were used to test data normality and homogeneity of variance, respectively. All analyses were completed using SPSS 20.0 (SPSS Inc., Chicago, IL, USA), and α-diversity of the bacterial microbes in the rhizocompartments of desert leguminous plants was analyzed using the Vegan package (R v3.1.1). To acquire the species taxonomy information corresponding to each OTU, the RDP Classifier algorithm (http://sourceforge.net/projects/rdp-classifier/) was used on the QIIME platform to comparatively analyze the representative sequences of OTUs, and to note species information of the different communities at various levels (kingdom, phylum, class, order, family, genus, and species). Alpha rarefaction curves were constructed using OTUs with 97% sequence similarity ($OTU_{97}$) for rarefaction analysis with mothur software, and rarefaction curves were constructed using sequencing data drawn and the OTU number

represented by them. The number of each sample randomly drawn was preset to start with 1, and calculated once for every increase of 50 until the rarefaction curves analyzing the OTU number of each sample all tended to be saturated on the whole. Differences in OTU composition among rhizocompartments, based on Weighted UniFrac distance, were analyzed using one-way analysis of similarity (ANOSIM) with 9,999 permutations. Principal co-ordinates analysis (PCoA) was used to evaluate overall similarities in microbial community structure based on the UniFrac distance. Hierarchical clustering of the samples based on Bray–Curtis dissimilarity was performed using QIIME. Indicator species analysis was performed using the multipat function of the indicspecies package in R 3.3.1 software. Mantel test was used to determine and quantify the major soil factors shaping microbial community structures. PCoA, ANOSIM, and Mantel test were implemented using the Vegan package in R 3.3.1 software [64]. Redundancy analysis (RDA) of the bacterial communities in the rhizocompartments was performed using CANOCO for Windows 4.5. We have done detrended correspondence analysis (DCA) on the species-sample data ($OTU_{97}$) and chosen the analysis method according to the size of the first axis in the Lengths of gradient part: if the gradient length exceeds 4.0, canonical correspondence analysis (CCA) is preferred; if the gradient is between 3.0–4.0, both RDA and CCA are suitable; if the gradient is shorter than 3.0, RDA is better than CCA. RDA was used to analyze the relationship between microorganism and environmental factors.

### Ethics statement

The study site does not contain any national park or other protected areas of land or sea. Environment Protection and Forestry Bureau of Yanchi County supervised the protection of wildlife and the environment. The location is not privately owned or protected, and the field studies did not involve endangered or protected species. No specific permits were required for the described field studies. For Yanchi Research Station was found by Beijing Forestry University and authorized by China government. The authorities and we authors confirm that the field studies did not involve endangered or protected species.

## Results

### Alpha rarefaction curve and α-diversity of microbial communities

Results from our analysis indicate that the diversity of root endophyte microbiomes was far lower than that of under-shrub soil bacterial microbial communities. Moreover, compared to various under-shrub soil samples, root sample rarefaction curves recorded a loose distribution in the saturated amplitude range and soil sample rarefaction curves (especially rhizosphere soil samples) recorded a more concentrated distribution in the saturated amplitude range. For most root samples, when the sequence number was close to 20,000 (X-axis) and the corresponding value range was 1000–1500 OTUs (Y-axis), the curve reached saturation (Fig 1A). For the under-shrub rhizosphere samples, the rarefaction curve reached saturation when the sequence number was close to 20,000 (X-axis) and the corresponding value range was 1500–3500 OTUs (Y-axis) (Fig 1B). For the root zone soil samples, the rarefaction curve reached saturation when the sequence number was close to 25,000 (X-axis) and the corresponding value range was 1500–3500 OTUs (Y-axis) (Fig 1C). For the bulk soil samples, the rarefaction curve reached saturation when the sequence number was close to 42,000 (X-axis) and the corresponding value range was 2000–4000 OTUs (Y-axis) (Fig 1D). For the convenience of further statistical analyses regrading sequencing depths of various sequencing samples, we listed the Good's coverage indices of various rhizocompartments based on more than 10,000 iterative computations in mothur (Fig 1). While the Good's coverage of all rhizocompartment samples (roots, rhizosphere soil, root zone soil, and inter-shrub bulk soil) was comparable, the diversity

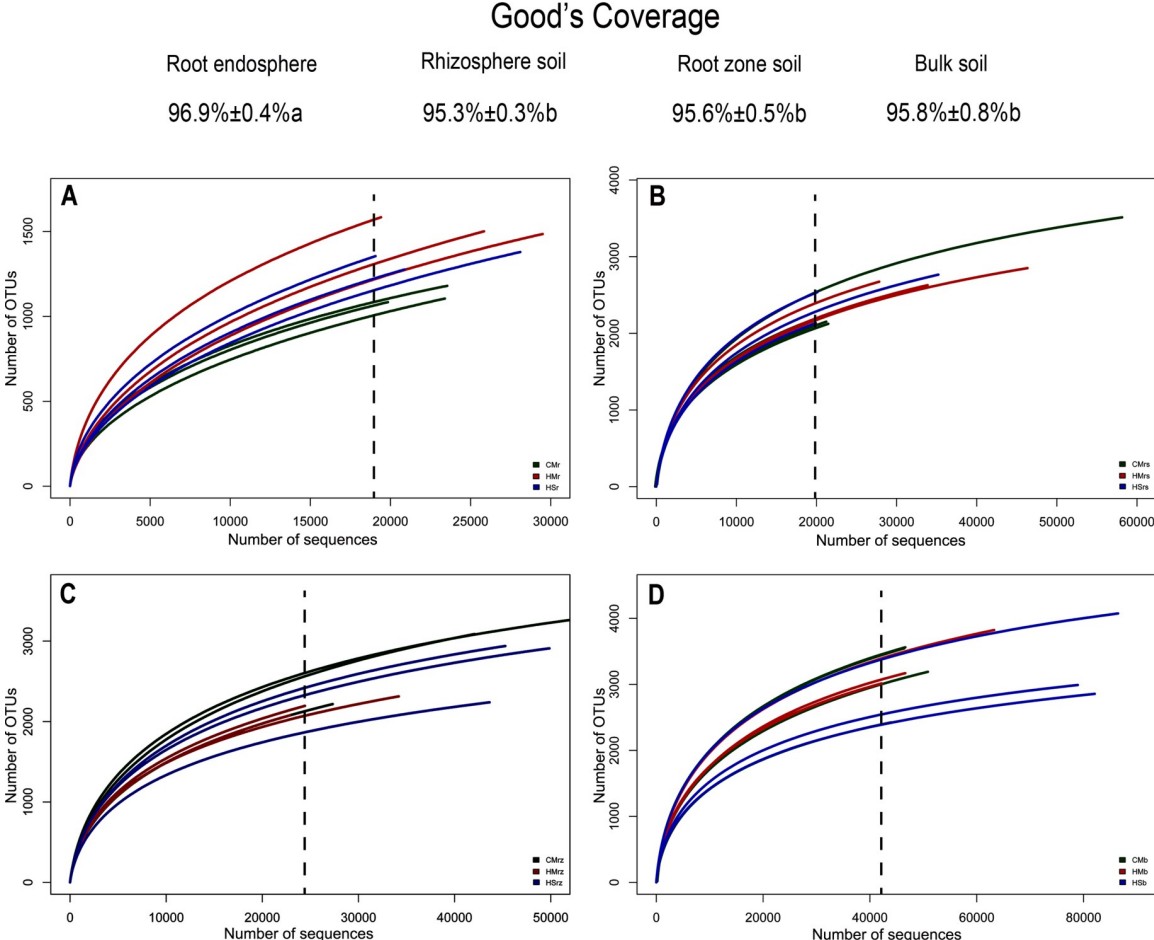

**Fig 1. Average Good's coverage estimates (%) and rarefaction curves.** Average Good's coverage estimates (%) represents the mean ± standard deviation of nine samples of each rhizocompartment type (three plants × three replicates); lowercase letters represent statistically significant differences within the 95% confidence interval ($P<0.05$). (A) root samples; (B) rhizosphere soil samples; (C) root zone soil samples; (D) inter-shrub bulk soil samples. CM: *Caragana microphylla*; HM: *Hedysarum mongolicum*; HS: *Hedysarum scoparium*. r: root; rs: rhizosphere soil; rz: root zone soil; b: inter-shrub bulk soil.

was not fully characterized due to insufficient sequencing depth. In this study, Good's coverage ranged from 95.3% to 96.9%; Good's coverage of the soil samples was significantly less than that of the root samples ($P < 0.05$).

Based on the OTU number, Chao1 bacterial species abundance index, and Shannon microbial diversity index, α-diversity analysis was conducted on the microbial diversity of various samples (Table 1). Results indicate that an obvious separation in α-diversity existed between the root samples of the three desert leguminous plants and soil samples, that the diversity indices of under-shrub soil samples were significantly higher than those of plant root samples ($P < 0.05$). Specifically, for the two *Hdysarum* species, the number of OTU, as well as their Chao1 and Shannon values of the samples collected in the rhizosphere soil were significantly higher than those of samples collected in the root and soil samples in the root zone ($P < 0.05$). By comparison, for the three desert leguminous plants, samples of the same rhizocompartment type all showed highly similar richness and diversity estimations; plant species and plant rhizocompartments jointly affect the alpha diversity of soil bacterial communities under and between shrubs, but rhizocompartments play a leading role (Table 1).

**Table 1. High-throughput genome sequencing results of the rhizocompartment samples of the three leguminous plants.**

| α-diversity index | | OTU | Chao1 | Shannon |
|---|---|---|---|---|
| Shrub species | Rhizocompartments | | | |
| CM | r | 1086±63 b | 1682.23±32.65 b | 7.15±0.33 b |
| | rs | 2300±224 a | 3244.72±306.22 a | 9.50±0.19 a |
| | rz | 2260±99 a | 3192.19±242.44 a | 9.33±0.33 a |
| | b | 2025.57±192 a | 3037.48±306.13 a | 9.33±0.22 a |
| HM | r | 1436±191 c | 2133.36±105.03 c | 7.34±0.99 c |
| | rs | 2340±148 a | 3098.14±117.83 a | 9.53±0.20 a |
| | rz | 2014±108 b | 2790.81±100.36 b | 9.08±0.05 b |
| | b | 2059±192 ab | 2842.44±127.15 b | 9.16±0.48 ab |
| HS | r | 1319±126 c | 2101.61±143.51 c | 7.24±0.54 c |
| | rs | 2374±161 a | 3309.04±130.52 a | 9.56±0.16 a |
| | rz | 1998±92 b | 3050.33±81.11 b | 9.10±0.07 b |
| | b | 2025.07±355 ab | 2863.53±150.19 b | 9.11±0.48 ab |
| $F_{shrub\ species}$ | | 0.57 ns | 0.14 ns | 0.10 ns |
| $F_{rhizocompartments}$ | | 47.72 | 30.64 | 71.05 |
| $F_{shrub\ species×rhizocompartments}$ | | 7.70 | 8.08 | 5.47 |

Data are presented as mean ± standard deviation, n = 3. CM: *Caragana microphylla*; HM: *Hedysarum mongolicum*; HS: *Hedysarum scoparium*.

r: root; rs: rhizosphere soil; rz: root zone soil; b: inter-shrub bulk soil.

Different letters signify significant differences within species among four rhizocompartments (P < 0.05). ns: no significant difference.

The last three rows represent the F values of the interactions between plant species/rhizocompartment types (shrub species×rhizocompartments).

## Beta diversity of microbial communities

We adopted two evolutionary phylogenetic levels (OTU and phylum) to assess the Beta diversity of microbial communities across different rhizocompartments. PCoA analysis was used to visualize the overall similarity among various rhizocompartment samples in the structures of bacterial communities, thereby comparing the compositions of microbial communities detected in various rhizocompartments. In addition, we also adopted algorithms describing the relationships and structures of community compositions to calculate inter-sample distances, i.e., performing hierarchical clustering analysis for verification purposes based on a Bray-Curtis distance matrix (Fig 2).

PCoA results (Fig 2A) indicated that, depending on the sources of the rhizocompartments of different desert leguminous plants (roots, rhizosphere soil, root-zone soil, and inter-shrub bulk soil), bacterial communities recorded a very strong clustering performance. When PCoA was based on the OTU level, PCoA1 and PCoA2 accounted for 53.9% and 10.26% of total variability, respectively. In addition, similar results were also obtained by grouping sources of various rhizocompartment samples and on hierarchical clustering based on a Bray–Curtis distance matrix at the phylum level, thereby verifying such clustering performance (Fig 2B). Hierarchical clustering analysis results indicated that root samples of the three desert leguminous plants were clustered according to rhizocompartment type; the other three rhizocompartments (rhizosphere soil, root zone soil, and inter-shrub bulk soil) differed from root samples and they did not cluster completely according to their respective rhizocompartment types (Fig 2B). To further verify the clustering performance of bacterial communities in various rhizocompartments in our PCoA results, ANOSIM was performed on samples from different rhizocompartments. As indicated by analysis results, there were significant differences among various rhizocompartmental types (*r* = 0.4953, *P* = 0.001) (Table 2 and S1 Fig). We found that the results of PCoA,

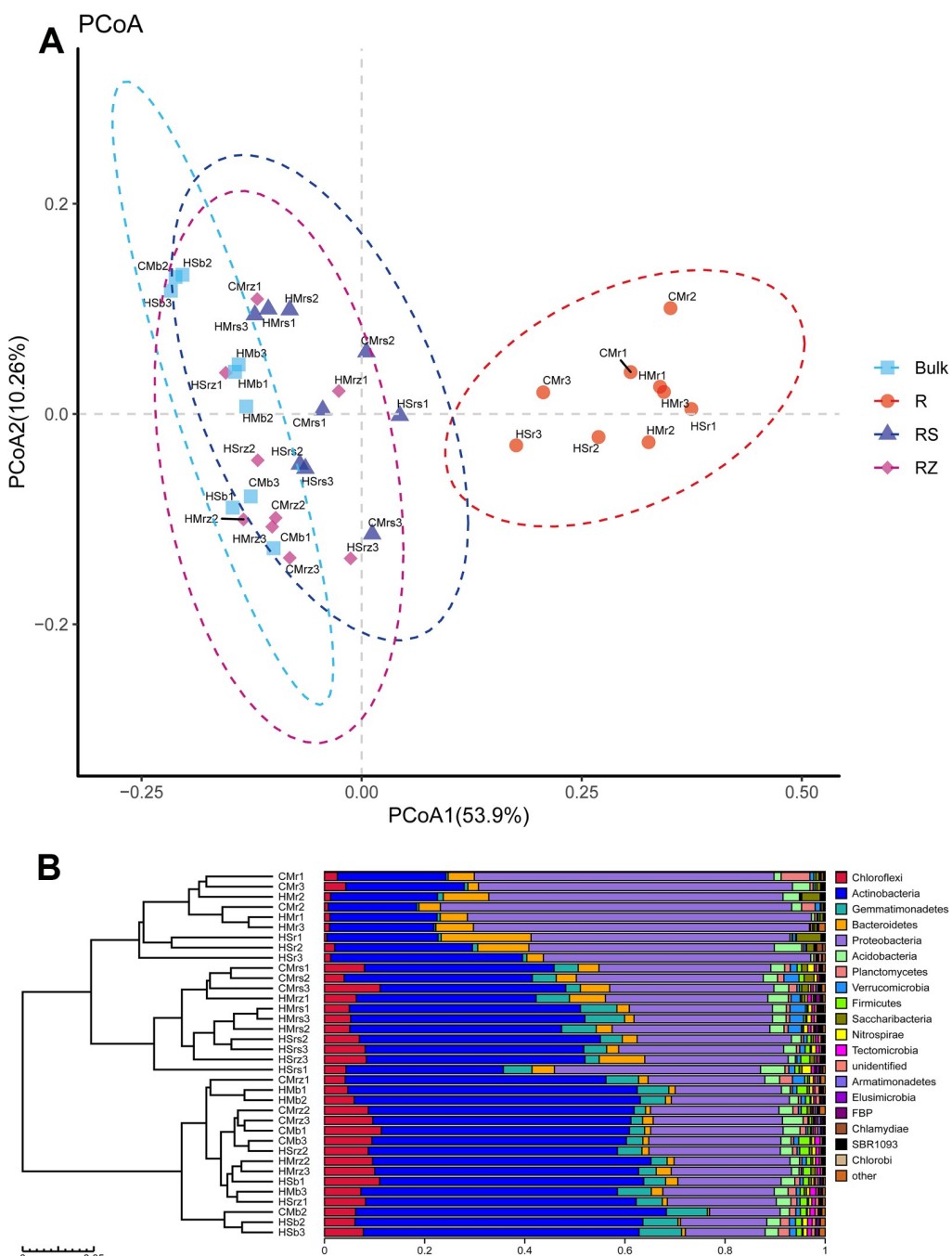

**Fig 2. Compositions of bacterial communities driven by rhizocompartments at OTU and taxonomic levels.** (A) PCoA on the compositions of bacterial microbes in the rhizocompartments of desert leguminous; (B) Hierarchical clustering analysis on samples based on Bray-Curtis distance matrix. CM: *Caragana microphylla*; HM: *Hedysarum mongolicum*; HS: *Hedysarum scoparium*. r: root; rs: rhizosphere soil; rz: root zone soil; b: inter-shrub bulk soil.

ANOSIM, and hierarchical clustering analysis were slightly different. PCoA and ANOSIM are methods based on UniFrac distance algorithm whereas hierarchical clustering analysis is based on Bray-Curtis distance algorithm. UniFrac distance algorithm could avoid errors in OTU clustering by reducing the influence of similar sequences.

**Table 2. Analysis of similarity (ANOSIM).**

| Phylogenetic level | Phylum | | OTU | |
|---|---|---|---|---|
| ANOSIM output | r | P-value | r | P-value |
| R vs. RS | 0.9499 | 0.001 | 0.9057 | 0.001 |
| R vs. RZ | 0.9952 | 0.001 | 0.9585 | 0.001 |
| RS vs. RZ | 0.4366 | 0.004 | 0.2541 | 0.023 |
| R vs. Bulk | 1 | 0.001 | 0.987 | 0.001 |
| RS vs. Bulk | 0.749 | 0.001 | 0.4784 | 0.001 |
| RZ vs. Bulk | 0.0706 | 0.137 | 0.1022 | 0.08 |
| Rhizocompartments | 0.6426 | 0.001 | 0.4953 | 0.001 |

Plant rhizocompartments effects on the microbial community structures were calculated using analysis of similarities (ANOSIM) based on the Weighted UniFrac distance metric (999 permutations).

R: root; RS: rhizosphere soil; RZ: root zone soil; Bulk: inter-shrub bulk soil.

Significance levels: $P \leq 0.05$; $P \leq 0.01$.

## Analysis of the differences in the structural compositions of major contributing bacterial taxa in various rhizocompartments

Differences in bacterial communities in the four rhizocompartments of desert leguminous plants at the phylum-order-genus levels were analyzed in-depth. At these taxonomic levels, ANOVA was used to assess the top ten ranked contributing phyla-orders-genera in terms of relative abundance percentage in the four rhizocompartments, respectively, so as to investigate the influence of rhizocompartment types on the structures and compositions of bacterial communities at various taxonomic levels (Figs 3 and 4 and S1 Table). In the four rhizocompartments of the three desert leguminous plants, bacterial microbes at the phylum level mainly included Proteobacteria, Actinobacteria, and Bacteroidetes (ten major contributing dominant bacterial phyla in total), accounting for 96%-99% of the total number of bacterial communities in the rhizocompartments (Fig 3 and S1 Table). The filtered major contributing bacterial phyla had significant differences in their relative abundances among the four rhizocompartments. Specifically, for Proteobacteria, the enrichment of *C. microphylla* and *H.scoparium* shrubs had the same trend: root > rhizosphere soil > root zone soil ($P < 0.01$), whereas the enrichment of Actinobacteria had the opposite trend ($P < 0.05$) (Fig 4 and S1 Table).

Under the three desert leguminous plant shrubs, bacterial communities at the order level mainly included Rhizobiales, Burkholderiales, and Sphingomonadales (ten major contributing bacterial orders in total), accounting for 12%-75% of the total number of bacterial communities in the various rhizocompartments (Fig 4 and S1 Table). Specifically, Rhizobiales was the dominant bacterial order with the highest relative abundance under the three leguminous plant shrubs, presenting a significant trend of enrichment towards roots in all of the four rhizocompartments under leguminous plant shrubs ($P<0.01$); Xanthomonadales also presented a significant trend of enrichment towards roots under leguminous plant shrubs ($P < 0.05$). Burkholderiales manifested a trend of enrichment towards roots under *H. mongolicum* shrubs ($P < 0.01$), and Rhodospirillales showed a trend of enrichment towards roots under *C. microphylla* shrubs ($P < 0.01$) and *H. scoparium* shrubs ($P < 0.02$). *Bradyrhizobium*, *Rhizobium*, and *Reyranella* were the bacterial genera with the highest relative abundances in roots under *C. microphylla*, *H. mongolicum*, and *H. scoparium* shrubs, respectively, with all bacterial genera presenting a significant trend of enrichment towards roots ($P < 0.01$). Although our results also indicated that *Rhizobium* presented an enrichment trend in *C. microphylla* roots ($P < 0.05$) and *Massilia* presented an enrichment trend in *H. mongolicum* roots ($P < 0.01$),

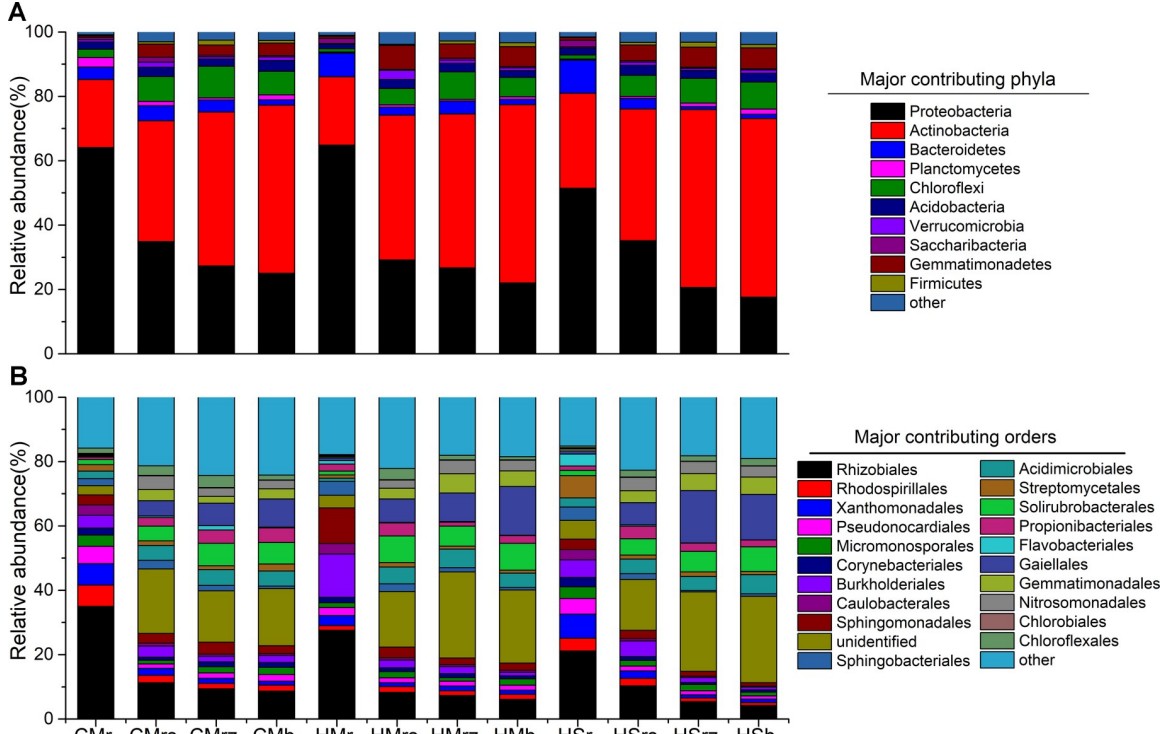

**Fig 3. Compositions of microbial communities in the four rhizocompartments of the three leguminous plants at phylum-order levels.** (A) The composition of microbial communities in the four rhizocompartments of the three leguminous plants at the phylum level; (B) The composition of microbial communities in the four rhizocompartments of the three leguminous plants at the order level. a and b list the top ten major contributing bacterial phyla and the top 20 major contributing bacterial orders, respectively, in terms of relative abundance percentage; the remaining contributors are represented by the category "other". CM: *Caragana microphylla*; HM: *Hedysarum mongolicum*; HS: *Hedysarum scoparium*. r: root; rs: rhizosphere soil; rz: root zone soil; b: inter-shrub bulk soil. The relative abundances of major contributing bacterial taxa in the four rhizocompartments of the three leguminous plants at phylum-order-genus levels and significant effects are listed in S1 Table.

some similar trends were also manifested in other major contributing bacterial genera in the four rhizocompartments (Fig 4 and S1 Table).

## Enrichment and filtration effects of specific bacterial taxa among rhizocompartments

Although bacterial communities in rhizocompartments originate from under-shrub soil, our results indicate that there are varying degrees of significant differences in the structures of microbial communities among the four rhizocompartments. Therefore, in order to generate data on the microbial species that cause significant differences in microbial communities between root/rhizosphere soil/root zone soil and inter-shrub bulk soil, we used the OTU number of inter-shrub bulk soil as a control measurement. We also introduced the Benjamini and Hochberg's algorithm to conduct significance of inter-group difference analysis on rhizocompartments and perform inter-group comparative analysis on OTUs ($P \leq 0.05$). Ultimately, we obtained the numbers of significantly enriched and significantly depleted OTUs of the three rhizocompartments relative to inter-shrub bulk soil. Compared to the bacterial communities of inter-shrub bulk soil, those of root/rhizosphere soil/root zone soil recorded 58/290/300 significantly enriched OTUs, and 568/340/321 significantly depleted OTUs, respectively. Based on Venn diagrams of the significantly enriched and depleted OTUs, root zone soil showed

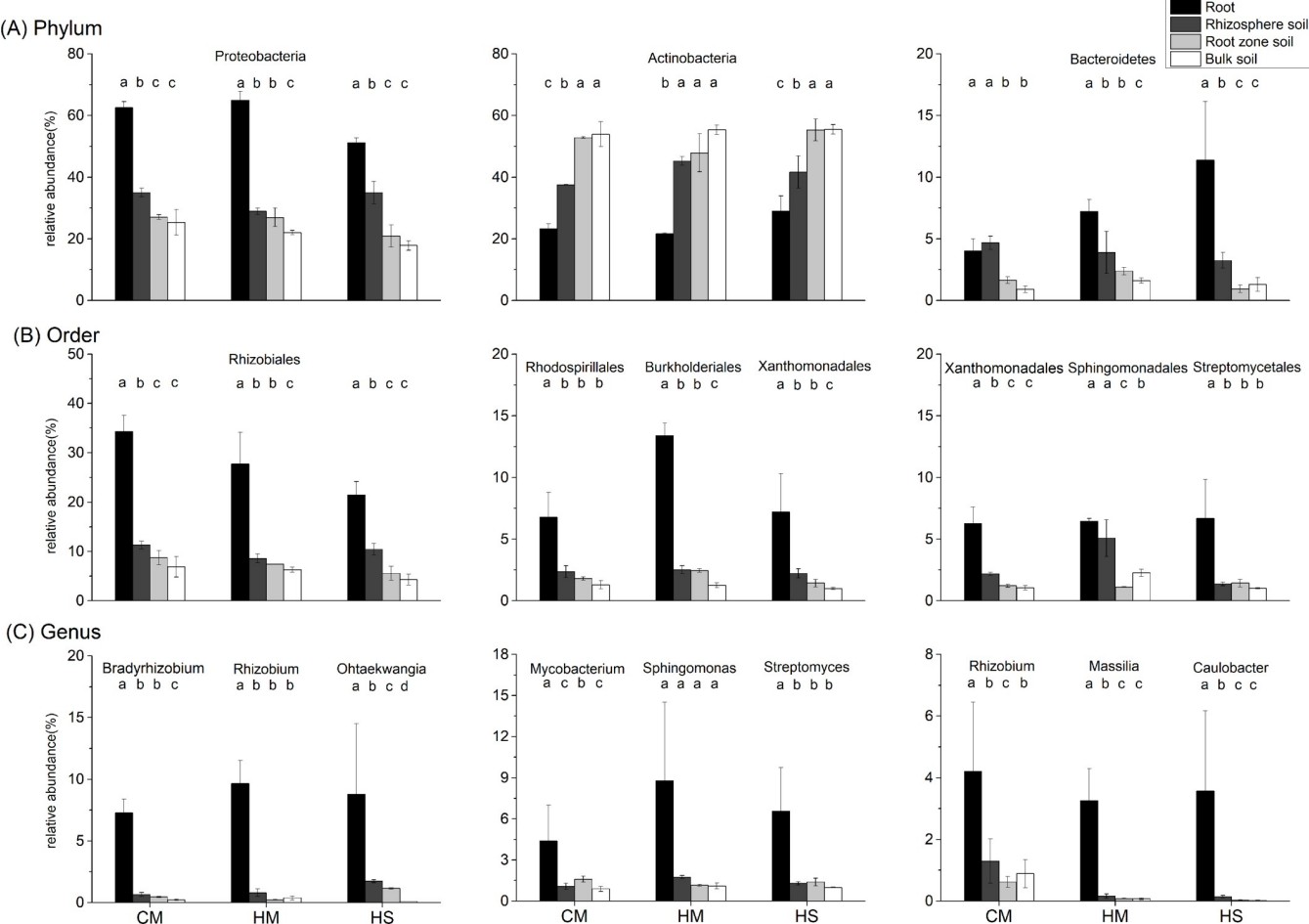

**Fig 4. Analysis of the significance of differences in the mean relative abundances (±SE) of major bacterial communities in the four rhizocompartments of the three leguminous plants at phylum-order-genus levels.** Group (A) provides the relative abundances of three major bacterial phyla; Group (B) provides the relative abundances of three major bacterial orders; Group (C) provides the relative abundances of three major bacterial genera. Different letters represent the presence of significant differences among the four rhizocompartments ($P < 0.05$). Black: root; dark gray: rhizosphere soil; light gray: root zone soil; white column: inter-shrub bulk soil. Data presentation: mean ± standard deviation, $n = 3$. CM: *Caragana microphylla*; HM: *Hedysarum mongolicum*; HS: *Hedysarum scoparium*.

smaller differences in the structures and compositions of bacterial communities when compared to inter-shrub bulk soil (Fig 5).

Enriched OTUs among various rhizocompartments all recorded varying degrees of overlapping. Specifically, among the 58 OTUs enriched in roots relative to inter-shrub bulk soil, 55/52 manifested consistent enrichment trends with OTUs from rhizosphere soil/root zone soil, respectively. Among the 290 OTUs enriched in rhizosphere soil, 209 manifested consistent enrichment trends with OTUs from root zone soil ($P \leq 0.05$). Fifty OTUs manifested an enrichment trend in all four rhizocompartment types set in this study, with root zone soil recording the largest number of significantly enriched OTUs (Fig 5A). We performed statistical analysis for ten relatively enriched dominant bacterial genera corresponding to OTUs. The analysis revealed that these genera in their bacterial microbial community structure of the rhizocompartment were relatively enriched ($P \leq 0.05$) (Fig 5C).

Among the 568 significant relatively depleted OTUs in roots, 338/320 presented consistent depletion trends as OTUs from rhizosphere soil/root zone soil. Among the 340 relatively

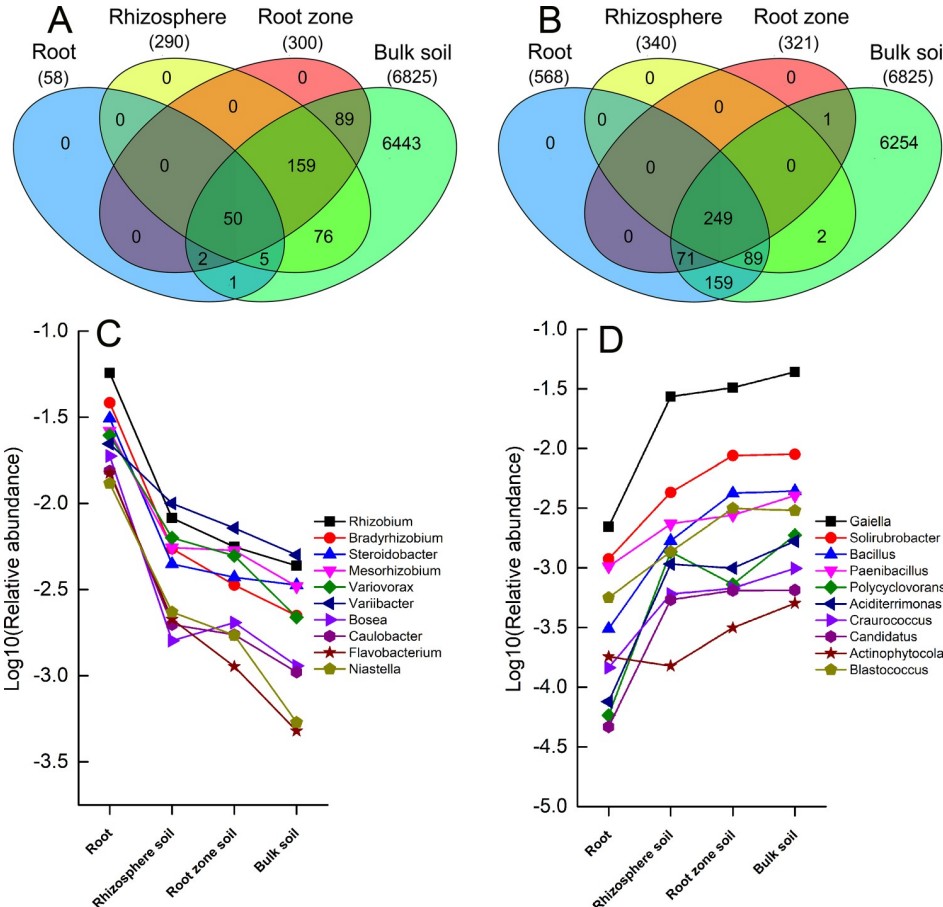

**Fig 5.** Venn diagrams of the significantly enriched (A) and depleted (B) OTUs in the other three rhizocompartments under leguminous plant shrubs compared to the inter-shrub bulk soil. (C) Bacterial genera corresponding to relatively enriched OTUs; (D) Bacterial genera corresponding to relatively depleted OTUs. The enrichment and filtration effects of OTUs among rhizocompartments are listed in S2 Table.

significantly depleted OTUs in rhizosphere soil, 249 presented consistent depletion trends as OTUs from root zone soil. Results indicate that the number of significantly depleted OTUs was small in root zone soil and highest in roots (Fig 5B). Similarly, statistical analysis on the ten relatively abundant dominant bacterial genera corresponding to the OTUs recorded a relative depletion effect among the four rhizocompartments. It shows that the proportion of these dominant bacterial genera in the rhizocompartment relatively depleted towards the roots ($P \leq 0.05$) (Fig 5D).

## Relationships between rhizocompartment bacterial communities and soil factors

In this experiment, soil samples were collected from the rhizosphere, root zone and inter-shrub bulk soil of three desert leguminous plants. All soil factors of rhizosphere and root zone soils, except for $NO_3^-$-N and TP under *C. microphylla* and the two *Hedysarum spp.*, recorded significant differences ($P < 0.05$). Apart from a slight deficiency in $NH_4^+$-N observed in rhizosphere soils under the three shrub species, values for all other rhizosphere soil nutrient indices were uniformly higher than those in the root zone or in the inter-shrub bulk soil. In three

cases, pH of the rhizosphere soil was lower than that of the root zone soil (Data of soil physico-chemical factors are listed in S3 Table).

Correlations between microbial community structure and soil physicochemical factors in rhizocompartments were calculated to identify abiotic factors that could cause variation in bacterial community diversity. Redundancy analysis (RDA) of bacterial communities in rhizo-compartments revealed that the samples were separated according to soil physicochemical factors in different rhizocompartment types (Fig 6). Among the four rhizocompartment bacterial communities, the microbiomes of root endophyte and rhizosphere soil were most strongly correlated to SWC and soil nutrient; communities in the root zone soil and inter-shrub bulk soil were most strongly correlated to soil pH and $NH_4^+$-N (Fig 6A). Members of the major contributing bacterial genera, which underwent hierarchical filtration and enrichment through inter-shrub bulk soil to roots by legume plants, were most strongly correlated to SWC and soil nutrients. The remainder of the major contributing bacterial genera, which were rela-tively depleted in roots compared with the other three rhizocmpartments, were mainly influ-enced by soil pH and $NH_4^+$-N (Fig 6B). Mantel test results revealed that soil pH, TN, SOC, and TP were significantly correlated with the microbial communities ($P < 0.05$) (Table 3).

## Discussion

### Structural difference of bacterial communities in rhizocompartments

When comparing root and soil samples, we observed that the rarefaction curves of OTUs recorded different saturation values and curve distributions upon reaching saturation. Rare-faction curve indicates a correlation between the differences in species abundance (OTUs) among the rhizocompartments of desert leguminous plants and the enrichment and filtration of specific probiotic bacteria by roots in the under-shrub soil bacterial communities under dif-ferent growth states of various sample plants. A study of niches in the roots of *Populus tremula* by Beckers et al. recorded similar rarefaction curve variations, indicating that uneven coloniza-tion of bacterial communities with root distribution may result in these findings [47]. Our

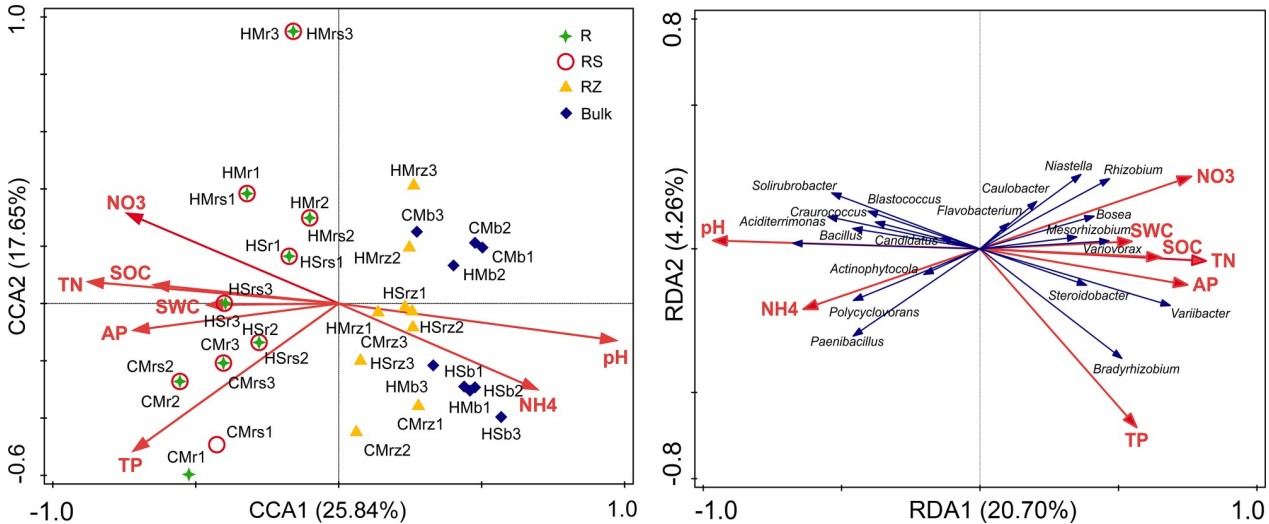

**Fig 6. Ordination plots of the results from the redundancy analysis (RDA) on the bacterial communities in rhizocompartments and soil factors.** (A) Correlation of soil factors to the samples of four rhizocompartment types under three leguminous plant shrubs. (B) Effects of soil factors on the major contributing bacterial taxa (20 filtered dominant genera in Fig 5C and 5D) in four rhizocompartments under three leguminous plant shrubs. CM: *Caragana microphylla*; HM: *Hedysarum mongolicum*; HS: *Hedysarum scoparium*. r: root; rs: rhizosphere soil; rz: root zone soil; b: inter-shrub bulk soil. Data of soil physicochemical factors are listed in S3 Table.

**Table 3. Correlation between bacterial community and soil properties as shown by Mantel test.**

| Soil properties | pH | TN | N-NH$_4^+$ | N-NO$_3^-$ | SOC | TP | AP | SWC |
|---|---|---|---|---|---|---|---|---|
| R | 0.2478 | 0.1413 | -0.051 | -0.01 | 0.1781 | 0.1333 | 0.1299 | 0.0573 |
| P-value | **0.002** | **0.047** | 0.693 | 0.522 | **0.036** | **0.041** | 0.089 | 0.224 |

Bold type indicates significant difference ($P < 0.05$)

data showed that the uneven distribution of the specific bacterial microbiomes that are colonized in the root system of each plant species among the rhizocompartments are the main reason for the difference of the rarefaction curves. Although derived nutrients (such as root exudates) and chemotaxis induction exist extensively in rhizosphere niches, plant-related bacteria must undergo intense competition before successfully migrating to and colonizing rhizocompartments [65]. Compared to the bacterial composition in soil among the rhizocompartments under shrubs, root endophytes must have many other properties to colonize the roots of host plants (such as the expression of chemotaxis-related genes, flagella, and the production of key enzymes for colonizing root cells) [66–68]. Bacteria that enter roots not only need to tolerate to stress factors caused by the innate immune system of host plants [66], but also have complex interactions with host plants to promote plant growth [67]. It was previously highlighted that plant-selected bacterial microbes in rhizosphere soil can enter root tissues and form endophyte microbiomes, and that their community compositions may differ from the composition of microbial communities in rhizosphere soil [69]. Lundberg et al. and Peiffer et al. recorded that root endophyte microbiomes are not purely random of bacterial microbial communities in rhizosphere soil, and that they are due to many complex factors, such as plant growth period, breed, and genotype [68, 70]. Bulgarelli et al. pointed out that the soil type determines the composition of bacterial communities in roots; however, the influence of host genotype on the community composition cannot be ignored [71].

In fact, for a), the bacteria associated with plants must be adapted to environmental variations (such as humidity, pH, and nutrient acquisition) in the micro-niche of rhizocompartments [72, 73], and only the highly competitive bacteria can successfully colonize the root system [33, 74]; and for b), the intricate interaction between soil-borne bacteria and the host plant's immune system [65, 67, 69, 70, 75] are dependent on the colonization ability and characteristics of endophytes [67, 68]. The structural differences of bacterial microbes in the rhizocompartments are demonstrated by the Alpha rarefaction curve (Fig 1), PCoA analysis and RDA analysis (Figs 2 and 6), and ANOVA of the relative abundances of major bacterial taxa among rhizocompartments (S1 Table) and ANOSIM of the structures of bacterial microbes in this study (S1 Fig). However, OTU number, Chao1 abundance, and Shannon diversity results (Fig 2) of rhizosphere soil, root zone soil, and bulk soil under desert leguminous plant shrubs were all higher than those of root endophyte microbiomes (Table 1).

## Drivers of the differentiation of bacterial communities under desert leguminous plant shrubs

Each rhizocompartment under the three leguminous plant shrubs were generally composed of Proteobacteria, Actinobacteria, and Bacteroidetes (Figs 3, 4 and 6 and S1 Table). This finding is consistent with the structure and composition of the dominant bacterial phyla reported in the relative abundance statistics by Sun et al. on the diversity of soil microbial communities under xeric shrubs in the Mu Us Desert [8]. The structure and composition of the major contributing bacterial phyla in root endophyte microbiomes of the three leguminous plants were

basically the same. Similar to findings on root and soil bacterial communities of *Arabidopsis* [76], *Acacia* [32], *Populus* [33], soybean, and alfalfa [51], results from our study indicate that a significant depletion in the relative abundance of Actinobacteria in bacterial microbial community structure in each rhizocompartment from the inter-shrub bulk soil to roots; Proteobacteria and Bacteroidetes in bacterial microbial community structure in each rhizocompartment showed enrichment towards the roots. It has been shown that the trend of under-shrub enrichment of Proteobacteria (mostly Alphaproteobacteria and Gammaproteobacteria) towards roots is positively correlated with the available carbon pool of soil, that Alphaproteobacteria are closely related to heterotrophic N-fixers in high-C plots, meaning that their presence can promote an increase of $NH_4^+$ pools [77], and that the under-shrub enrichment of Bacteroidetes is attributable to their ability to rapidly utilize organic matter in the soil [78]. In addition, Fierer et al. have confirmed that, compared to the oligotrophic of Acidobacteria and Beta-Proteobacteria in soils under shrubs, Bacteroidetes have better symbiotic nutritional properties, and that the under-shrub enrichment of Bacteroidetes is not only limited by soil organic carbon content [79], it is also affected by other factors such as soil texture [80]. Findings by Nemergut et al., Lauber et al. and Van Horn et al. also highlighted that the relative abundance of Actinobacteria is mainly affected by soil pH [77, 81, 82]. At the order level, Rhizobiales was the major contributing bacterial order with the highest relative abundance percentages in the roots of desert leguminous plants. In addition, under *H. mongolicum* shrubs, Burkholderiales were significantly enriched in the roots. Deng et al. and Moulin et al. reported that Burkholderiales enriched in roots may have the potential to promote plant growth [83, 84]. It has also been shown that differences between niches are possibly caused by the distribution of nutrient resources in various niches [85], and other soil abiotic factors such as soil pH, soil moisture content, and soil nutrient availability [86, 87].

No nodule samples were collected and analysed in this study; however, the degree of microbial diversity in the roots of desert leguminous plants is vast, the dominant locations of *Bradyrhizobium* and *Rhizobium* in the roots of *C. microphylla* and *H. mongolicum* are also predictable (Figs 4 and 6, and S1 Table) as bacteria of these two genera are known to have symbiotic rhizobia, and they are closely related to the symbiotic nitrogen fixation of leguminous plants [51, 88]. However, in rhizocompartments under *H. scoparium* shrubs, neither of these two bacterial genera had significant relative abundance percentages or enrichment significance. In *H. scoparium* roots, *Ohtaekwangia* was the bacterial genus with the highest relative abundance percentage and enrichment significance (Fig 4 and S1 Table). Results from previous studies examining the succession of bacterial communities in the rhizosphere soil of corn highlighted *Ohtaekwangia* to have a dominant role in the early growth phase of corn, to be readily able to degrade organic matter as substrates, and to be copiotrophic and fast-growing bacteria [85, 89]. *Rhizobium*, *Bradyrhizobium*, and *Bosea*, as well as other dominant bacteria detected in the roots of desert leguminous plants in this study, also exist in the roots of *Acacia salicina* and *A. stenophylla* (Mimosaceae) in southeastern Australia [90, 91], and in the roots of wheat growing in volcanic ash soil in southern Chile as microsymbionts [92]. The significant enrichment of the major contributing bacterial taxa under leguminous plant shrubs across the four rhizocompartments are possibly related to the hierarchical filtration of probiotic bacteria by leguminous plants [51].

Results for the mantel test identified pH, TN, SOC, and TP to be the major factors correlated with bacterial community structure under desert leguminous plant shrubs (Table 3). As far as the plant species cited above are concerned, the major contributing bacterial communities in their rhizocompartments all have enriched quantities of bacteria affiliated to the Phylum Proteobacteria at various taxonomic levels. According to our results, Proteobacteria was the dominant bacterial phylum in the compositions of bacterial communities in the four

rhizocompartments of the three desert leguminous plants at the phylum level. For the order level, Rhizobiales was the dominant bacterial order (Figs 3–6 and S1 Table). For example, the specific bacterial microbiomes *Bosea*, *Rhizobium*, and *Mesorhizobium*, which are abundant in the rhizosphere soil, are mainly correlated with $NO_3^-$ and TN (Table 3 and Fig 6). Research shows that only a few microorganisms can absorb and assimilate nitrate nitrogen in soil. When nitrate nitrogen and ammonium nitrogen are present at the same time, ammonium nitrogen can inhibit the nitrate nitrogen absorption of microorganisms, because the assimilation of nitrate nitrogen requires energy consumption [93]. From the rhizocompartment to the root system, the interaction between plant roots and soil microorganisms is gradually strengthened, but the absorption preference of nitrogen-fixing microorganisms for ammonium nitrogen resulted in fixation and accumulated of ammonium nitrogen [94, 95]. This probably explains our observation that the ammonium nitrogen content of rhizosphere soil to become significantly lower than nitrate nitrogen content (S3 Table). Therefore, nitrogen in rhizosphere soil ($NO_3^-$ and TN) may differences in certain bacterial microbiomes (especially for *Bosea* and *Mesorhizobium*) (Table 3 and Fig 6). In the compositions of endophyte microbiomes in host plants, the vast overlapping of specific microbiomes suggests that endophytic capacity (effective enriched colonization) and plant rhizocompartments (such as nutrient availability/variability. pH, and habitat suitability) are all retained for specific bacterial microbiomes (Fig 5 and S3 Table). It is possible that the significant hierarchical enrichment and depletion trend of specific bacterial microbiomes in the roots of host plants is not just a passive process, and that it depends on the active filtration of bacterial microbiomes by host plants or the opportunistic colonization of some bacteria in suitable niches [33, 96, 97].

## Conclusion

This study revealed the diversity characteristics of bacterial microbial communities in the rhizocompartments of three desert leguminous plants (*C. microphylla*, *H. scoparium*, and *H. mongolicum*), and discovered that the specific bacterial microbiomes in under-shrub soil microbial communities had a significant hierarchical enrichment effect among rhizocompartments, and were filtered into roots. In this case, the structure composition of root endophytic communities significantly differed from the bacterial communities in the other three rhizocompartments; the bacterial species abundance and microbial diversity of the root endophytic communities were significantly less than that of the bacterial communities in the other three rhizocompartments. In addition, our data also verified that the rhizocompartments of under desert leguminous plant shrubs had a significant differentiation effect for the specific bacterial microbiomes enriched and filtered by host plants, and the formation of the root endophytic communities was influenced by the rhizocompartments and plant species. This study provided data to support the identification of plant-related dominant bacteria and the preparation of soil improvement inocula in the future.

## Supporting information

**S1 Fig. Plant rhizocompartments effects on the microbial community structures were calculated using analysis of similarities (ANOSIM) based on the Weighted UniFrac distance metric (999 permutations).**
(TIF)

**S1 Table. Analysis of the significance of differences in the mean relative abundances (±SE) of major contributing bacterial taxa in the four rhizocompartments of the three leguminous plants at phylum-order-genus levels.**
(XLSX)

**S2 Table. The enrichment and filtration effects of OTUs among rhizocompartments.**
(XLSX)

**S3 Table. Soil physicochemical properties of rhizospheres and root zones under the two shrub species (n = 3).**
(XLSX)

# Acknowledgments

We would like to thank the staff of the Yanchi Research Station, especially Genzhu Wang, Shijun Liu, Yangui Qiao, Yuxuan Bai for their help with experimenting and sampling in the field.

# Author Contributions

**Conceptualization:** Ziyuan Zhou, Guodong Ding.

**Data curation:** Ziyuan Zhou, Minghan Yu, Guanglei Gao, Yingying He.

**Formal analysis:** Ziyuan Zhou, Minghan Yu.

**Funding acquisition:** Guodong Ding.

**Methodology:** Ziyuan Zhou, Minghan Yu, Guanglei Gao, Yingying He.

**Project administration:** Minghan Yu, Guodong Ding, Guanglei Gao.

**Visualization:** Ziyuan Zhou.

**Writing – original draft:** Ziyuan Zhou, Minghan Yu.

**Writing – review & editing:** Minghan Yu, Guodong Ding, Guanglei Gao.

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
