## [Decision Letter · Decision Letter 0]

12 Apr 2020

PONE-D-20-03936

Diversity and Structural Variability of Bacterial Microbial Communities in Rhizocompartments of Desert Leguminous Plants

PLOS ONE

Dear Dr. Ding,

Thank you for submitting your manuscript to PLOS ONE. After careful consideration, we feel that it has merit but does not fully meet PLOS ONE’s publication criteria as it currently stands. Therefore, we invite you to submit a revised version of the manuscript that addresses the points raised during the review process.

While all four reviewers thought the study interesting, they all noted substantial issues with the data analysis and interpretation, the lack of clarity and logical organization, and the poor quality of presentation of the manuscript.

We would appreciate receiving your revised manuscript by May 27 2020 11:59PM. To enhance the reproducibility of your results, we recommend that if applicable you deposit your laboratory protocols in protocols.io, where a protocol can be assigned its own identifier (DOI) such that it can be cited independently in the future. For instructions see: http://journals.plos.org/plosone/s/submission-guidelines#loc-laboratory-protocols

We look forward to receiving your revised manuscript.

Kind regards,

Brenda A Wilson, Ph.D.

Academic Editor

PLOS ONE

Journal Requirements:

Reviewers' comments:

Reviewer's Responses to Questions

**Comments to the Author**

1. Is the manuscript technically sound, and do the data support the conclusions?

Reviewer #1: Partly

Reviewer #2: Partly

Reviewer #3: Yes

Reviewer #4: Partly

2. Has the statistical analysis been performed appropriately and rigorously? 

Reviewer #1: Yes

Reviewer #2: I Don't Know

Reviewer #3: Yes

Reviewer #4: No

3. Have the authors made all data underlying the findings in their manuscript fully available?

Reviewer #1: No

Reviewer #2: No

Reviewer #3: Yes

Reviewer #4: Yes

4. Is the manuscript presented in an intelligible fashion and written in standard English?

Reviewer #1: Yes

Reviewer #2: Yes

Reviewer #3: Yes

Reviewer #4: Yes

5. Review Comments to the Author

Reviewer #1: This work describes the microbial communities associated with three desert legume species, including two Hedysarum spp. and Caragana microphylla, as well as the various physicochemical properties of the soils form which they were isolated. An improvement of this work over similar studies referenced by the authors (i.e. Sun et al.) is the separation of under-shrub soils into “rhizocompartments” which include root tissue and associated soils in the rhizosphere. The methods used to isolate the samples, as well as the statistical analyses and soil property quantification approaches, are somewhat well-described but would be strengthened by adding some additional details as suggested below.

My main concern about this manuscript is that the interpretation of the statistical approaches, and of the perceived microbial community diversity, is inadequate. There are many cases in the manuscript (some of which I have highlighted below) whether the authors claim that “all” samples follow a certain trend or meet a certain significance level, when only the majority of samples do. Similarly, the authors’ methods for collecting DNA samples likely exclude any microbes that are not tightly associated to soil or roots, and the possible effects of this selection on the results are not discussed. These overstatements of the strength of the results make it difficult to have confidence in the authors’ interpretations.

Specific comments:

Introduction: The introduction is quite general, and provides little information about the specific plant species studied and what is already known about their associated microbiota. For example, it is mentioned that these species are legumes, yet it is not mentioned whether they engage in nitrogen-fixing symbioses with bacteria (which not all legumes do), and if so which bacterial symbionts are compatible, although there are many studies of this topic for Hedysarum spp. available in the literature. Some of this information is provided in the discussion, but giving more specific background for the chosen plant systems earlier in the paper will help readers contextualize the results.

Methods: It is unclear to me how the “inter-shrub bulk soil” was defined – how far away were the soil samples from the shrubs? The methods described for collection of the microbiota also do not seem to be optimized for maximum DNA sample recovery. Some questions that I had about the sample prep are:

• Was the soil DNA extraction kit also used for the root samples, or was a different extraction kit that is optimized for plant tissues used?

• How were the roots processed before extraction to break up the plant tissues (e.g. in a mortar and pestle or with enzyme treatment) – and were nodules included with the other root tissues?

• Why was the buffer that was used for ultrasonication of the roots, which likely includes many epiphytes that are more stably attached to roots as well as very fine soil particles, not added to the rhizosphere DNA sample?

• Did the authors test to see whether a 15 minute spin at 3200g is sufficient to collect the majority of microbes in the rhizosphere sample? In this method, anything that is not stably attached to the soil particles will be lost, since many rhizobia, for example, can’t be pelleted at such a low speed. I suspect the authors are losing quite a few microbes in their washes.

While it’s true that all extractions result in some sample loss, and many studies have followed similar protocols as the authors of this study, I recommend that the authors re-define their rhizocompartments based on the way the samples were processed to improve clarity. For example, the “rhizosphere soil” microbiota is perhaps better to define initially as “microbes strongly adhered to coarse root-associated soil” (although I think it is fine to use“rhizosphere soil” after this sample category is introduced).

Results: In virtually every main and supplemental, letters are used to represent groups of samples that are not statistically significant from one another, but specific p-values for comparisons between members of the “a”, “b”, “c”, etc. groups are never provided. The sample sizes are also not explicitly mentioned in the legends. This information should be added to the results, otherwise the strength of the results is very difficult to evaluate.

Additional comments:

• By “under-shrub” samples, do you mean the root, root zone and rhizosphere samples collectively? Please define.

• For Fig. 1, it would be helpful if each panel had different line widths or line types for each of the host species. Additionally, two different legends are provided for this figure (lines 251-260 and again on lines 261-265), which is confusing.

• In Fig. S1, a p-value is shown, but again I am not clear what this represents – which samples are being compared?

• Regarding the results presented in Table 1, in lines 271-273 of the manuscript the authors claim “Specifically, OTU number, Chao1 index, and Shannon index of rhizosphere soil samples for the two Hedysarum L. plants were all significantly higher than those for root samples and root zone soil samples, as well as those of inter-shrub bulk soil samples (P<0.05).” However, the rhizosphere soil samples are only significantly different from the inter-shrub bulk soils using the Chao1 index but not for the OTU number and Shannon index. Similarly, on lines 275-276, the authors claim “no significant effect of plant species on bacterial richness and diversity was observed” yet the root zone samples of species CM are indeed different from the root zone species of HM and HS using all three methods.

• In lines 217-319, the authors state “Specifically, Proteobacteria presented a significant step-by-step enrichment trend in the order of root>rhizosphere soil>root zone soil (P<0.01) under the three desert leguminous plant shrubs; Actinobacteria, Gemmatimonadetes presented a contrary trend (P<0.05)”. However, in Table S1, there is no difference in the Proteobacteria or Actinobacteria abundance between rhizosphere and root zone soils for sample HM, and there is no different between rhizosphere and root zone soils for Gemmatimonadetes for any of the plant species. There are similar overstatements of the results presented throughout the manuscript.

• For Fig. 5A-B, the raw data used to generate these diagrams should be provided in the supplement – or, if it is already provided in one of the tables, make it clear which one.

• For Fig. 6/Table S2, why pick the top ten rather than choose OTUs to define as “dominant” based on their relative abundance (e.g. anything comprising over X% of the total OTUs for all four samples)? I am also not sure what meaningful new information is added in Fig. 6 that is not already provided in Fig. 5C-D, other than including a specific representative species name. I suggest moving this tree to the supplement and shortening the corresponding section of the results.

• In Fig. 7, graphs are not labeled as (a) or (b).

Reviewer #2: The manuscript titled as "Diversity and Structural Variability of Bacterial Microbial Communities in Rhizocompartments of Desert Leguminous Plants" compared microbiome from four compartments, and found root microbiomes had higher structural variabilities than the other 3 types of microbiomes.

Major concern: The comparison of four compartments is interesting. However, ANOSIM table is missing, therefore it is hard for the reviewer to determine whether the conclusion is solid for the structural variability. Please present the table which should contain the R and P-value. To avoid confusion, a summary akin to table 2 in this other paper https://link.springer.com/article/10.1186/s40168-017-0241-2 is the one missing.

Minor comments:

1. Even though the written English is intelligible, a thorough proof-reading is highly recommended.

2. The resolution of the figures are poor, and the reviewer could not download tables as it shows to be corrupted. Therefore, it was hard to determine the clustering in Figure 2B. Figure 4 letters for significance are hard to distinguish.

Reviewer #3: In general, the topic of the Diversity and Structural Variability of Bacterial Microbial Communities in Rhizocompartments of Desert Leguminous Plants is an important area to develop better understanding. Much of the actual work performed on the manuscript and general analysis was adequate. However, as pointed below, the manuscript contains some problems, improper data interpretation, data presentation, and discussion. These need to be revised carefully.

Comments for abstract: I am not quite sure the format of abstract published by Plos One, however, the results and conclusion in the abstract need to include meaningful data. For example, line 26 “had significant effects on …”, what does this “significant effects” mean? It could be positive or negative effect, and “specific bacterial microbiomes”, what is this “specific bacterial microbiomes” mean? And line 35-36, “while soil physicochemical factors have a significant influence”, what is this significant influence? I would recommend the author rewrite this abstract.

Line 63: there is an error, the meaning of this sentence is not clear, “improve soil” or “improve soil quality/health”?

Line 68: “participate in and control many processes of soil ecosystems” need to rewrite, here might be some errors. Probably “participate in and control many soil ecosystem functioning processes” is better to understand.

Line 71: citation needed here.

Line 81: “30,000 prokaryotes per gram of root” does this mean 30,000 species or this is the number? If the author means how many species in each gram of root soil, please indicate it here.

Line 148: “corresponding” might be better than “according”

Line 251-255: Don’t need figure cations here. This is a duplicate as below line 261-265

Table 1: Error in the first row, third column, the title for OTU.

Line 272: use “both” instead of “all”

Line 278-285: there is no need to explain what analysis you did for your sample in the results part, you may explain in materials and methods. I would suggest the author consider remove it. Just state what significant results you get from your sequencing data.

Line 303: use “r” instead of “R”

Line 305: there is a mistake in this sentence

Line 311: figure 4A and 4B are duplicate in figure 3, why does the author create these two graphs? If the author want to show results at phylum, order, and genus levels, please create another graph at genus level use sequencing data as showed in figure 3.

Line 313: format error at the end of the line

Line 344-345: Error in this sentence, please rewrite.

Before discussion in Table 2, first column, use “P-value” instead of “P”

Line 482: need to indicate which figure represent this result, so people are easier to read and understand the manuscript

Line 481-489: I understand the author would like to explain and discuss why bacterial communities in roots have a higher structural variability compared to soil bacterial communities. However, the author only provided several research findings from Gottel et al, Bulgarelli et al, Lundberg et al. and Peiffer et al, I don’t see any evidence that can support the author’s point here. Instead, if the author could provide support on line 490-491, then it will be easier to understand.

Line 574: Error in this sentence “formed had low richness and diversity”.

In supporting materials: S4 table title “S4 Table. Soil physicochemical properties of rhizospheres and root zones under the two shrub species (n = 3).” I think there are three shrub species. And the unit for TP is not correct.

Reviewer #4: Dear authors and editors,

In the manuscript “Diversity and Structural Variability of Bacterial Microbial Communities in Rhizocompartments of Desert Leguminous Plants” the authors aimed to characterize the microbial communities found in, on, near, and further from the roots of several shrub species in a particular desert habitat to determine if shrubs influence the microbial communities. The authors calculate metrics of alpha diversity and find that root endophytic samples were least diverse, but that root zone soil is more diverse than between-shrub soil. – i.e. shrubs influence microbial communities. The sampling and sequencing methods are appropriate for this goal, although I have noted a few questions that came up in line comments. I commend the authors on these aspects.

However, I have a few major concerns about the analyses, and one about the writing and organization. Briefly, while some analyses were good, I found others redundant, some poor quality, and a key analysis to support a conclusion seemingly missing. I found the text on data analysis in the methods greatly lacking in detail, with some misplaced to the results, and not all of the discussion and conclusions are well-linked to the data. Note that lack of clarity could partly explain some of my concerns about the analyses. While my concerns may seem grievous, because the manner of data collection and sequencing appears fundamentally valid, I believe the authors could address all my concerns.

Major Comments:

I believe the authors wished to quantify how variable communities from certain sample types were WITHIN that type. This is what I think the authors mean by “structural variations” and similar language throughout the manuscript. Conclusions about this are reported (“we found that the formation of bacterial microbial communities in root was a higher variability process”), but I see no clear test of this. For example, species turnover (beta diversity) between samples within a type appears to not be calculated, or it isn't explained. There are various appropriate calculations of beta diversity the authors could calculate with the data they have. Instead, two redundant metrics identifying compositional change ACROSS sample types are employed. Both PCA and hierarchical clustering of Bray-curtis distance are used, and both show that endophytic communities differ from the other sample types, but that the remaining sample types and all species are similar in species composition to each other (though I wonder if there are subtle differences, as I note below). These are useful analyses, but not sufficient to support the conclusion quoted above.

Likewise, the authors use several methods to ask which species differ across sample types (e.g. which species drive the result above). First, they could have discussed which taxa contribute to the axes in the PCA they already ran, but instead chose to use totally different analyses. They cut the dataset to more abundant groups (not sure exactly how, methods are sparse, but this is repeated across cases when grouping by phyla, order, or genus), then individually tested changes in relative abundance of each group across sample types. This ignores a major constraint – 16s rDNA is relative abundance, so changes in one group are NOT independent of others (when one increases in relative abundance, others must therefore decrease, even if absolute abundance in these others in the field has not changed). Therefore, taxa-by-taxa tests for significant trends do not make sense. Instead, I suspect the third analysis (if it had been more fully explained) might do just this: it appears to me from the text that the metastats packages looks for groups of OTUs with correlated changes in abundance, i.e. acknowledging that each change OTU abundance is not independent from the others. I would recommend the authors develop the results of this test more (which taxa are in correlated groups?), or use the correlated changes in OTUs from the PCA, and skip the other tests. Lastly, the authors define conditions for a core microbiome, and again test where it is enriched – with the circular result that the most abundant taxa from a set of samples are enriched in that sample. I may be missing something in the methods (see below), but if not, this circular analysis is not very helpful and should be removed from the manuscript, along with figure 6.

The authors investigate how soil physiocochemical factors differ across sample types. This analysis largely made sense. The authors also checked to see if some bacterial groups were correlated with these factors – my suggestion here is to use the full dataset rather than the core microbiome only – the core are the set of microbes that are stable across samples, and so the LEAST likely to be influenced by soil properties.

My last major comment is about the organization of writing and clarity of logic. Much data analysis text is both insufficient and found in the results instead of the methods. The data analysis section is not co-linear with how the tests were run or how the results were presented. The discussion section also varies in quality. Some places appropriately reference both the results and the literature to support conclusions (the first two paragraphs after the heading “Drivers of the differentiation of bacterial communities under desert leguminous plant shrubs” are great, for example). But the rest of the discussion is missing those key aspects. Given the places where the writing is is clear and does make sense (see referenced paragraphs in discussion, and the rest of the methods is much easier to follow), I expect the authors can fix this.

The data dryad link doesn't work for me, though I believe the authors that they have deposited their data, and expect there is a simple typo. Analysis scripts should be made available along with the data.

I have included line-referenced comments below, since I took the time to note them down when reading. Given the sweeping nature of the changes I recommended above, I do not know how useful these specifics will be to the authors or editors.

Intro

Line 56: It is a shortcut to say “The nitrogen-fixing ability of leguminous plants”, as it is associated microbes that fix nitrogen. The authors are clearly aware of this, so I assume this is a typo.

Lines 53-60: This paragraph seems less important towards motivating the study than the other points. The authors might cut this and enhance other points.

Line 66: Here and elsewhere, I think the authors may be accidentally mis-using “adaptability”, or they have not explained fully. Adaptability of desert vegetation would be if desert plants had a greater capacity to evolve (genetically) in response to selection pressures than other plants. There could be a literature stream on this of which I am not aware. However, I think the authors actually mean to communicate that desert vegetation has a high tolerance of extreme abiotic conditions.

Line 71: “believed” maybe isn't quite the right word. No one believes they are completely co-transmitted like a true genome, and I don't think that's what the authors mean here?

Line 72: This is an example of where specifics would help the writing. It's not totally clear what's meant by “The microbiome not only provides functional assistance and support to host plants” unless the reader has background knowledge already.

Lines 77-79 could use specificity and citations.

Line 81-83: Claims like this are made at various places in the manuscript. Sometimes (like here), there seems to be an unsupported degree of agency assigned to the plant. Elsewhere (559-562), a similar statement reflects more of what I have seen supported in the literature, and is also more informative.

Line 87: I don't understand: “core bacterial microbiomes closely associated with host plants dominate the degree of variability of the entire bacterial community across different niche”

Methods

Line 123: The sampling description is largely clear to the reader. Totals would enhance this clarity.

Line 130: Please describe somewhere in this section how tools were cleaned/sterilized across samples.

Line 131: For bulk soil samples, is this from a zone where there are no roots? If not, were there species of non-focal shrubs nearby, and if so, is there anything the authors can report on how this varied or did not vary across sampling locations?

Line 133: It’s my impression that most studies use much smaller and younger roots because there is more biological activity and more living microbes in newer tissue. A note on why you selected this size would be good.

Lins 125-1333: Root, root zone and rhizosphere seem to be sampled from the exact same place, whereas bulk soil is sampled from a different location. There should be a comment in the results/discussion about how this could or would not have affected your results.

Line 137: Not sure what “interval” means in this context.

Line 155: I infer roots are not analyzed for physicochemical properties, but the reader could use this explicitly stated.

Line 168: What restrictions? After reading the results, I think it makes more sense to simply drop the root samples and associated sequences from the analysis of soil physicochemical properties? The authors may disagree with me on this point.

Line 181: The selection of primers makes sense as a practical decision. However, as it isn’t considered best for capturing diversity of bacteria, the authors should add some discussion of how this could/should not have affected results.

Line 189: I note QIIME2 is available, and has better analytical tools (e.g. amplicon sequence variants vs operational taxonomic units). For this study, I think version 1.8 is fine --I just want to bring this to the authors' notice, in case they want to try it out.

Line 198: I don't know what “the sequence directions were adjusted” means.

“Statistics analysis” section: This needs way more detail. I won't go line by line at all. I will recommend the authors consider using UNIFRAC distance rather than Bray-Curtis. It could help overcome OTU clustering errors by downweighting the influence of similar sequences, and would distinguish this analysis from the PCA (unless this is a UNIFRAC-based PCA, as I think QIIME can do?).

Results: This is not a complete line-by-line for this section, since I assume much text will change, but I have a few thoughts. I generally have not commented on the figures, as I expect they may change or that revised methods/results text might make them easier to interpret.

It is normal to report various stats on sequencing: e.g. total reads or bp sequenced, total reads post quality filters, range of read-depth per sample, # of OTUs, # of OTUs assigned taxonomy identifiers etc. etc.

Figure 1: I do not think the rarefaction curves look flat – i.e. not saturated. Did the authors use a method to estimated the number of OTUs they might have seen at saturation? I think they may have, but that descriptive detail is lacking? I also do not see referenced sequencing depth statistics (mean and SE). I think some text in the caption or following text includes an error: Good's coverage is non-overlapping, which instead suggests that some groups of samples are not sequenced as completely as others, but the conclusion is opposite.

The phrasing in lines 272-273 seems out of sync with reported numbers in Table 1. The words suggest alphaRS>alphaB (not supported by data), but I think the authors mean to say: alphaB>alphaRZ, similar to how alphaRS>alphaRZ (supported by data).

Line 282: “and exploring the main influencing factors driving differences in micro-community compositions”. PCA describes a pattern, not causality. Also on the PCA, why only report 2 axes? Do other axes not distinguish other groups (e.g. sites, species) or do they not explain sufficient variance?

Figure S1: There is no y-axis. Is it Bray-curtis distance? Why are bulk samples separated in panel a? No mention is made of the logic for the different test groupings in the methods.

Just in case: In plotting & analysis software, sometimes multiple “unknown” taxa from QIIME may be grouped together because they share the name “unknown” and not because they are the same OTU or even similar enough to be in the same order/phyla. They looked grouped in figures, because there is a very high abundance taxa with unknown identity (unusual but not impossible). This could be intentional for figures, but for analyses it would not make sense to group different unknown taxa into one taxa.

Line 359: I do not know what “we used the OTU number of inter-shrub bulk soil as a control” means in this analysis.

Line 361: The related figure (5) seems to only compare rhizocompartments, not OTUs? I don't understand.

Figure 5: I don’t understand the Venn diagram figure or the analysis underlying it. How can 80 OTUs from the rootzone be enriched relative to the rootzone (in the root zone and bulk soil overlap)?

Line 417: What species indicator analysis? I didn't see this mentioned anywhere else...

Table 2: what aspect of communities are soil properties correlated with? I couldn’t see it in the methods.

Discussion: This is not a careful line by line commenting.

The first two paragraphs here are largely conclusions and relevant literature, but no references to the specific results that support conclusions. Instead, the third paragraph is a list of relevant tests without specifics. See my major comment for where I think the authors did a much better job and compare the two.

Line 468: “Our results indicate that variations in specific core bacterial microbiomes in massive

endophytes colonizing roots of various plant species” I do not understand this part, and so do not understand the whole sentence.

Lines 470-473: Do you have results supporting this claim? I did not see or understand them if so.

Line 479: “must” is not a good word here, and elsewhere. Not all endophytes promote growth.

Below illustrates my confusion on “structural variability”:

Lines 490- 491: “In summary, the colonization of endophytes in roots and the subsequent formation of a community with a relatively stable structural composition seem to constitute a highly variable process.” If the structure is stable, then where is the evidence for variability?

Line 510: Typo in order of treatments?

516-519: This text implies Sphingobacteriia is the only copiotrophic member of Bacteroidetes. It would surprise me if that were the conclusion of Fierer et al. I’m wondering if that’s truly what’s meant?

521-523: This is a bit of a cliff-hanger. Does pH explain the trend of Actinobacteria?

Line 525: What does “degree of significance in roots mean”? – (that’s how I read this.)

Line 528: Why mention of aromatic compounds? Are they relevant? The reason is missing if so.

Line 546-548: I am struggling to understand how this last sentence links in with Rhizobium & the data from the present study.

Paragraph starting on 549: I think this paragraph was meant to discuss how soil/niche properties might explain which microbes occur in which rhizocompartments. There are results that speak to this, but this paragraph doesn’t clearly reference those results and is instead repetitive of the previous two paragraphs.

562-564: Is this from the results (if so, which result specifically?) Or is this meaning to reference other work?

Line 570: I think the authors mean to say here that members of core microbiome of the root endophyte zone had a hierarchical enrichment. I think there is evidence for this. But as written this conclusion is confusing.

Line 573-574: The first clause I understand and see how the results support. I cannot say the same for the second clause.

Line 577: “each rhizocompartment represented a unique niche of bacterial communities”. Unless I missed something, this is not the case. Instead, the PCA and Bray-Curtis results suggest all but endophytic communities are similar.

Line 578-580. Clause 1 is supported by the data. For clause 2, I do not understand what the authors mean, exactly, and this is the first mention we've had in the discussion about restoration.

6. PLOS authors have the option to publish the peer review history of their article (what does this mean?). If published, this will include your full peer review and any attached files.

Reviewer #1: No

Reviewer #2: No

Reviewer #3: No

Reviewer #4: No

---

## [Author Response · Author response to Decision Letter 0]

10 Jun 2020

Dear reviewers：

Thanks to the reviewers for sparing time to go through the manuscript, highlighting very important issues and providing helpful comments and valuable suggestions to improve the manuscript. We have revised the manuscript seriously and carefully according to the reviewer’s comments and suggestions. Specific changes could be found in the revised manuscript now submitted. We hope it could offset the shortcomings in the original manuscript. More details and point-to-point responses to the reviewer’s comments were uploaded as a separate file, and labeled 'Response to Reviewers'.

---

## [Decision Letter · Decision Letter 1]

30 Jun 2020

PONE-D-20-03936R1

Diversity and Structural Variability of Bacterial Microbial Communities in Rhizocompartments of Desert Leguminous Plants

PLOS ONE

Dear Dr. Ding,

Thank you for submitting your manuscript to PLOS ONE. After careful consideration, we feel that it has merit but does not fully meet PLOS ONE’s publication criteria as it currently stands. Therefore, we invite you to submit a revised version of the manuscript that addresses the points raised during the review process.

While overall improved, there remain a number of issues to be addressed by reviewers 2, 3 and 4. In particular, reviewer 4 did not feel that their concerns were adequately addressed. In reviewing the revised manuscript and the reviewer comments, I must agree with this reviewer, especially with regard to over interpretation of some of the data, lack of clarity, and circular arguments. I believe all of the issues raised are reasonable, and I believe are sincerely meant to improve the quality of the manuscript. Since there is general consensus that this is a potentially interesting report with information of benefit to the community, I am willing to allow another attempt at addressing the issues raised by the reviewers.

We look forward to receiving your revised manuscript.

Kind regards,

Brenda A Wilson, Ph.D.

Academic Editor

PLOS ONE

Reviewers' comments:

Reviewer's Responses to Questions

**Comments to the Author**

1. If the authors have adequately addressed your comments raised in a previous round of review and you feel that this manuscript is now acceptable for publication, you may indicate that here to bypass the “Comments to the Author” section, enter your conflict of interest statement in the “Confidential to Editor” section, and submit your "Accept" recommendation.

Reviewer #1: All comments have been addressed

Reviewer #2: All comments have been addressed

Reviewer #3: All comments have been addressed

Reviewer #4: No

2. Is the manuscript technically sound, and do the data support the conclusions?

Reviewer #1: Yes

Reviewer #2: Yes

Reviewer #3: Yes

Reviewer #4: No

3. Has the statistical analysis been performed appropriately and rigorously? 

Reviewer #1: Yes

Reviewer #2: Yes

Reviewer #3: Yes

Reviewer #4: Yes

4. Have the authors made all data underlying the findings in their manuscript fully available?

Reviewer #1: Yes

Reviewer #2: Yes

Reviewer #3: Yes

Reviewer #4: Yes

5. Is the manuscript presented in an intelligible fashion and written in standard English?

Reviewer #1: Yes

Reviewer #2: No

Reviewer #3: Yes

Reviewer #4: No

6. Review Comments to the Author

Reviewer #1: (No Response)

Reviewer #2: The manuscript has improved in this current version and addressed my concerns about data analysis and presentation. However, this reviewer still believes that the writing needs to be improved by a scientific editor. Since certain wordings are still unchanged, here are some examples from version 1 that could be improved, but still unchanged in version 2:

1. L319 "display" should be "visualize"

2. L465 "divided" should be "separated" or "grouped"

3. L478 "interval" should be "value"?

4. In L492 and other places in the article: "migration" may not be appropriate in this context. The microbiomes are snap shots of what the compositions are. There is no data to show that bacteria moved as the term "migration" implies.

5. L500: "a few" is probably better as "plant-selected" or "specific"

6. L528 "soya bean" should be "soybean"

7. L529 "record" should be "exhibit"

8. L543 "contrary" should be "opposite"

9. L554 "position" should be "location"

10. L558 "prominent" should be "significant"

In version 2, L23 "while" should be "Moreover"?

These are just some obvious wordings I can pick up readily since this reviewer does not have time to edit the paper. There are subtle differences between words when they both translated to be the same in another language. Yet the actual meanings are very different. Please be aware of this and have a scientific editor help your overall writing.

Reviewer #3: Minor comments: Line 538: Bacteroidetes does not need Italic, Make sure it is consistent with rest of the phylum name.

Reviewer #4: Dear authors and editors,

As I reviewed the manuscript previously, I have primarily evaluated changes and responses to my previous comments.

The authors of this manuscript have made a number of substantial and commendable improvements to their manuscript. The text changes to the introduction have better set up the study. The methods have improved in the clarity of text, the use of UNIFRAC distance, the better treatment of alpha diversity metrics, the use of Benjamini and Hochberg's algorithm to handle multiple testing issues (though this was only a small part of my original critique), and the use of the full community data for tests of community composition correlation to soil variables. Further, interpretation of alpha diversity metrics in the results has also improved, and several of the major issues I raised have been addressed or become more minor issues.

However, 2 major concerns I detailed on initial review (I was reviewer #4) were not addressed adequately, and one not at all. Importantly, this means that the discussion and conclusions still make statements that are not supported, and in one case a conclusion that was not even tested. This is why I marked "no" on question 2. A number of my minor comments were also not fully addressed. I think this manuscript requires more substantial revision. Generally, I strongly suggest that the authors focus on rephrasing their conclusions and discussion to fit what they have actually observed. The data as analyzed are useful, there is no need to over-interpret.

Major points:

1) I still see a conclusion about how variable communities from certain sample types were WITHIN that type (“the formation of bacterial microbial communities in root was a higher variability process under the joint influence of rhizocompartment habitats and host plants”, line 601), but I still see no clear test of the elements of this and there are several issues. First, changes in communities correlated to proximity to the plant do not indicate agency of the plant or even causation (correlation is also misinterpreted as causation in the discussion of correlated changes in microbial communities and soil physicochemical properties, see below), Second, I struggle to see where “higher variability” comes from. If I can find any result that the conclusions might be derived from, it is figure S1. If “higher variability” is such a key result, the rationale and results should be presented in the main text. Yet, I don't think ANOSIM is designed for this test, and is particularly weak on this point. “The anosim function can confound the differences between groups and dispersion within groups and the results can be difficult to interpret.” (quoted from package `vegan' manual for R, see Warton et al. 2012, full citation below). Alternatively, a few places in the text lead me to believe that the authors may instead be misusing “highly variable process” to describe the opposite pattern – a deterministic pattern (e.g. see line 514). The other conclusions drawn from ANOSIM with respect to differences between groups of samples (e.g. roots vs bulk soil as reported in Table 3) appear generally sound to me, and they agree with other methods.

2) I made a previous comment about circular analyses, which relates to R1's comment about similar data being presented in multiple figures. This is still true about this manuscript. However, the authors respond that they replicated the analyses of another paper in order to provide direct comparisons (Beckers et al 2017 and Wang et al 2019). I understand that it can be useful to compare across studies, even when previous methodologies have underlying issues. I now instead recommend that the authors state this reasoning in the methods, and results. They should also provide statements that remind readers what can and cannot be inferred from changes in proportion data, and tone down their inferences. Specifically, despite decreasing in relative abundance, a particular bacterial taxonomic group might actually be increasing in total abundance. Also, when some groups are relatively higher in abundance, the logical requirement is that others will be less relatively abundant, but only one of them must change in absolute abundance to create this result in relative abundance. Changes cannot be assigned with certainty to particular taxa. Likewise, when selecting taxa that are most relatively abundant across samples that include more and less complex communities, taxa relatively enriched in less complex communities (here, roots) will therefore be more “relatively abundant” when pooling across groups. This is why first limiting to relatively abundant taxa across samples and then asking if relatively abundant taxa are enriched in roots is circular.

3) Much data analysis text is still insufficient and still found in the results instead of the methods (for an example, see paragraph starting line 431, but this occurs elsewhere too). However, this problem has been reduced. Some paragraphs of the discussion still don't make sense (in combination with my first major point, this is largely why I marked "no" on question 5), however this problem is now lessened.

Other comments that were insufficiently addressed, or are based on edited material or a new understanding of previous material:

1) “Adaptability” which refers to the adaptive, evolutionary potiential of a population, is still used to refer to the concept that individuals may be able to tolerate a wider range of environments (in many places throughout). The authors should use “wide tolerance” or “plasticity” in most of these cases.

2) My hologenome comment was mis-interpreted. I took issue with the intent, not the vocabulary. See, for example: Douglas, Angela E., and John H. Werren. "Holes in the hologenome: why host-microbe symbioses are not holobionts." MBio 7.2 (2016).

3) I do not understand how the results on lines 284 -292 were generated. These curves still don't look like they are leveling off, although I agree with the authors that the method to draw them & relevant methods text seems improved. I suggest the authors acknowledge that diversity in any one sample might not be fully characterized due to sequencing depth – and again, tone down conclusions to fit this reality. Low power doesn’t make research useless, but it is very important to fully communicate the certainty of conclusions. Note this also relates to material in the results and discussion.

4) Table 1: The origin of letters denoting statistical differences are not fully explained, and therefore, some presented results are confusing. For example, it seems implied that in Shannon diversity, 9.10±0.07 b is different than 9.11±0.48 a, but I can easily see that standard errors overlap. So what are a and b telling me? This occurs at several places in the table. Conversely, bulk soil often appears to be not significantly less diverse than rhizosphere soil, so I struggle to believe the claim that there are significant “peaks” in diversity in the rhizosphere soil (as claimed on 309-310).

5) I note PCA, ANOSIM, and hierarchical clustering give slightly different results, but this is not discussed.

6) I still do not understand why the analyses or comparisons in Figure 5 are undertaken. This may be related to my point about circularity of proportion data and the Becker paper.

7) On line 487 the authors state about their results that “In fact, the enrichment of bacterial communities towards rhizosphere soil mainly derived from the following sources: (i) carbon-containing primary and secondary metabolites (root exudates) in plant rhizospheres; and (ii) allelochemicals released by plants through allelopathy, which induces chemotaxis in oil microbes [40]” but there is no experimental result documenting this, or method testing it. I recommend the authors remove this and similar conclusions that are unsupported.

8) Paragraph beginning 522: This paragraph is hard to parse. It may need to be simplified, especially given that relative enrichment and absolute abundance are not necessarily linked, see my major comments.

9) Paragraph beginning line 553: This is useful content, but readers should be reminded that nodules were explicitly avoided in sampling, so the lack of enrichment of nodule taxa is somewhat unsurprising.

10) line 506: This language ascribes more agency to plants, than is actually supported by Bulgarelli. – likewise on 503, a better interpretation is that communities are not purely random. They could be deterministically opportunistic. I think this is a grammar error.

Citations

Warton, D.I., Wright, T.W., Wang, Y. 2012. Distance-based multivariate analyses confound location and dispersion effects. Methods in Ecology and Evolution, 3, 89–101

Jari Oksanen, F. Guillaume Blanchet, Michael Friendly, Roeland Kindt, Pierre Legendre, Dan McGlinn, Peter R. Minchin, R. B. O'Hara, Gavin L. Simpson, Peter Solymos, M. Henry H. Stevens, Eduard Szoecs and Helene Wagner (2019). vegan: Community Ecology Package. R package version 2.5-5. https://CRAN.R-project.org/package=vegan

7. PLOS authors have the option to publish the peer review history of their article (what does this mean?). If published, this will include your full peer review and any attached files.

Reviewer #1: No

Reviewer #2: No

Reviewer #3: No

Reviewer #4: No

---

## [Author Response · Author response to Decision Letter 1]

14 Aug 2020

Dear reviewers：

We are very grateful to the reviewers for taking the time to read the manuscript, pointing out problems of crucial importance, and offering helpful comments and valuable opinions about how to improve the manuscript after the first round of revision since the initial submission. We have submitted a new version of the manuscript and figures in accordance with the recommendations of the editor. We have addressed the comments raised by the reviewers, the letter was uploaded as a separate file, and labeled 'Response to Reviewers'. A marked-up copy of our manuscript that highlights changes made to the original version, and labeled 'Revised Manuscript with Track Changes'.

---

## [Decision Letter · Decision Letter 2]

14 Sep 2020

PONE-D-20-03936R2

Diversity and Structural Differences of Bacterial Microbial Communities in Rhizocompartments of Desert Leguminous Plants

PLOS ONE

Dear Dr. Ding,

Thank you for submitting your manuscript to PLOS ONE. After careful consideration, we feel that it has merit but does not fully meet PLOS ONE’s publication criteria as it currently stands. Therefore, we invite you to submit a revised version of the manuscript that addresses the points raised during the review process.

Reviewer 4 still has some minor issues that should be addressed.

We look forward to receiving your revised manuscript.

Kind regards,

Brenda A Wilson, Ph.D.

Academic Editor

PLOS ONE

Additional Editor Comments (if provided):

The reviewers note the substantial improvement of this revision. Reviewer 4 still has some good suggested edits for the authors that would further improve the manuscript. Please address these.

Reviewers' comments:

Reviewer's Responses to Questions

**Comments to the Author**

1. If the authors have adequately addressed your comments raised in a previous round of review and you feel that this manuscript is now acceptable for publication, you may indicate that here to bypass the “Comments to the Author” section, enter your conflict of interest statement in the “Confidential to Editor” section, and submit your "Accept" recommendation.

Reviewer #2: All comments have been addressed

Reviewer #4: (No Response)

2. Is the manuscript technically sound, and do the data support the conclusions?

Reviewer #2: Yes

Reviewer #4: Yes

3. Has the statistical analysis been performed appropriately and rigorously? 

Reviewer #2: Yes

Reviewer #4: Yes

4. Have the authors made all data underlying the findings in their manuscript fully available?

Reviewer #2: Yes

Reviewer #4: Yes

5. Is the manuscript presented in an intelligible fashion and written in standard English?

Reviewer #2: Yes

Reviewer #4: Yes

6. Review Comments to the Author

Reviewer #2: (No Response)

Reviewer #4: Dear editor and authors,

The authors have made careful revisions that have improved the clarity and quality of the manuscript. I think the conclusions drawn from the data and analyses are now largely sound (with minor exceptions, see first point), and that the study represents a valuable contribution.

I was the original R4, and R4 the last time around. I write with respect to how my comments on the previous submitted version (revision 1) have been addressed in this version (revision 2). Every change I suggest now represents a minor change.

In response to my first major point, the authors have clarified text and results and I am largely satisfied with the response. Only this portion “correlation is also misinterpreted as causation in the discussion of correlated changes in microbial communities and soil physicochemical properties,” remains a point of concern to me. The following wording changes would be sufficient to correct it:

Line 437 “Effects of soil factors on the” should be “Correlation of soil factors to the”

Line 448 “were mainly influenced by” should be “were most strongly correlated to” – same for line 449, line 451, and line 453.

Line 539: “affecting” should be “correlated with” and likewise Line 547: “affected by” should be “correlated with”

Line 554: “This results in” should be “This probably explains our observation that”

Line 556: “more markedly affects” should be “may differences in”

In response to my second major point – we are going to have to agree to disagree about what constitutes circularity in an analysis. However, the authors have removed all the conclusions that I thought were erroneous that hinged on the circular analysis. So I am satisfied.

In response to my third major point, I think both readability and distribution of methods text between methods/results have improved sufficiently.

In response to minor points--

1) there are still a few instances of “adapt” that aren't right. I highlighted all of them below.

2 & 3) Good responses, I am satisfied with these changes.

4) I am satisfied. It would (optionally) add even more clarity to change “Different letters signify significant differences among four rhizocompartments” in the Table 1 caption to “Different letters signify significant differences within species among four rhizocompartments”.

5) The changes are good.

6) Sorry, still don't understand panels A and B in Fig 5. Specifically, from reading the caption, I interpret that panels A and B show the number of taxa in the other samples that are enriched relative to root zone soil. Taxa in enriched in two other sample types, would then be summed in the overlap of corresponding colors. Yet numbers appear in zones that overlap with root zone sample. How can root zone be enriched relative to itself? i.e. in A, the overlap between rhizosphere, root zone, and bulk soil has the number 159.

7)-10) no further comments, changes seem good.

Quick wording changes to make meaning appropriate/clear.

Line 42: “adaptable” should be “tolerant”

Line 82: “adaptations” should be “tolerance”

Line 86: “differential variations” should be “differentiation”

Line 96: same as line 86.

Line 99: “variability” should be “differentiation”

Line 104: I would delete the rest of this sentence after “rhizocompartments” as there is no test of it. However, the conclusions are fine, so leaving this as a hypothesis that remains untested may be what the authors prefer. I would leave this up to the authors.

Line 127: “plasticity” should be “tolerance”

Line 190: this phrase might be unclear “were most closely interact with the physical and chemical properties”. I think the authors mean that because the root communities are correlated to physicochemical soil properties near the root, it suggests that near-root soil properties might influence them. If I have interepreted this correctly than it should be fine to leave as is.

Line 226: Should specify which databases for which sequencing data.

Table 3 appears inside the caption of Figure 6 (line 437).

Figure 6: Panel A says “CCA” – I believe this should be RDA? Or was a canonical correspondence analysis performed?--if so should be in methods near line 261. (note just “CCA” is confusing, because a canonical correlation analysis is a totally different procedure).

Line 473 “adapt” should probably be “tolerate” – although evolution may sometimes be involved

Line 481: “karyotype distribution” should probably be “community composition”

Line 532: “degradable” should be “able to degrade”

*note to authors. The meaning is clear in most places even where there are grammar issues, so I did not flag this in my review – however there are indeed grammar issues. If you paid for your english editing service, you might want to consider a different provider next time.

7. PLOS authors have the option to publish the peer review history of their article (what does this mean?). If published, this will include your full peer review and any attached files.

Reviewer #2: No

Reviewer #4: No

---

## [Author Response · Author response to Decision Letter 2]

28 Sep 2020

September 29, 2020

Dear reviewers：

We are very grateful to the reviewers for taking the time to read the manuscript, pointing out problems of crucial importance, and offering helpful comments and valuable opinions about how to improve the manuscript after the second round of revision since the initial submission. We have submitted a new version of the manuscript and figures in accordance with the recommendations of the editor. We have addressed the comments raised by the reviewers, the letter was uploaded as a separate file, and labeled 'Response to Reviewers'. A marked-up copy of our manuscript that highlights changes made to the original version, and labeled 'Revised Manuscript with Track Changes'.

---

## [Editor Report · Decision Letter 3]

8 Oct 2020

Diversity and Structural Differences of Bacterial Microbial Communities in Rhizocompartments of Desert Leguminous Plants

PONE-D-20-03936R3

Dear Dr. Ding,

We’re pleased to inform you that your manuscript has been judged scientifically suitable for publication and will be formally accepted for publication once it meets all outstanding technical requirements.

Kind regards,

Brenda A Wilson, Ph.D.

Academic Editor

PLOS ONE
---

## [Editor Report · Acceptance letter]

3 Dec 2020

PONE-D-20-03936R3 

Diversity and Structural Differences of Bacterial Microbial Communities in Rhizocompartments of Desert Leguminous Plants 

Dear Dr. Ding:

I'm pleased to inform you that your manuscript has been deemed suitable for publication in PLOS ONE. Congratulations! Your manuscript is now with our production department. 

Kind regards, 

on behalf of

Dr. Brenda A Wilson 

Academic Editor

PLOS ONE